# ARCHITECTURE-AGNOSTIC TEST-TIME ADAPTATION VIA BACKPROP-FREE EMBEDDING ALIGNMENT

**Xiao Ma[1], Young D. Kwon[2], Pan Zhou[1], Dong Ma[3,*]**

[1]Singapore Management University, Singapore
[2]Samsung AI Center–Cambridge, Cambridge, United Kingdom
[3]University of Cambridge, Cambridge, United Kingdom
`xiaoma.2022@phdcs.smu.edu.sg, yd.kwon@samsung.com`
`panzhou@smu.edu.sg, dm878@cam.ac.uk`

## ABSTRACT

Test-Time Adaptation (TTA) adapts a deployed model during online inference to mitigate the impact of domain shift. While achieving strong accuracy, most existing methods rely on backpropagation, which is memory and computation intensive, making them unsuitable for resource-constrained devices. Recent attempts to reduce this overhead often suffer from high latency or are tied to specific architectures such as ViT-only or CNN-only. In this work, we revisit domain shift from an embedding perspective. Our analysis reveals that domain shift induces three distinct structural changes in the embedding space: translation (mean shift), scaling (variance shift), and rotation (covariance shift). Based on this insight, we propose Progressive Embedding Alignment (PEA), a backpropagation-free and architecture-agnostic TTA approach. By applying a novel covariance alignment procedure at each intermediate layer, PEA efficiently corrects the embedding distortions with only two forward passes. Extensive experiments demonstrate that PEA achieves state-of-the-art performance in both accuracy and efficiency, while also proving versatile across different architectures including ViTs and CNNs. Code is released at https://github.com/TheMaXiao/PEA_TTA.

## 1 INTRODUCTION

Deep neural networks (DNNs) have achieved remarkable success across a wide range of computer vision tasks (Pouyanfar et al., 2018). However, their performance often degrades significantly under distribution shifts between the training data and unseen test data - a challenge that frequently arises in real-world and real-time applications (Koh et al., 2021; Sun et al., 2022). To address this limitation, DNNs must be able to adapt effectively to such shifts. Test-time adaptation (TTA) (Liang et al., 2025) has recently emerged as a promising paradigm, enabling a pretrained model to be fine-tuned on-the-fly using unlabeled test batches as they arrive during inference. By continually adjusting to new data distributions, TTA mitigates the performance degradation caused by domain shifts and enhances the robustness of deployed models.

Most mainstream TTA approaches rely on either pseudo-labeling or entropy minimization. Pseudo-labeling (Wang et al., 2022; Marsden et al., 2024; Lee & Chang, 2024) is a self-supervised strategy that assigns provisional labels to the current test batch and updates the model based on these label estimations. In contrast, entropy minimization (Wang et al., 2020; Niu et al., 2022; 2023) is an unsupervised method that encourages the model to produce more confident predictions directly from unlabeled data. Despite their effectiveness, both approaches suffer from a fundamental drawback: *they depend on backpropagation*. Specifically, they require backward passes and gradient storage across multiple layers during adaptation, which introduces substantial computational and memory overhead. This reliance makes them unsuitable for deployment in resource-constrained settings, such as edge devices or real-time applications. Recent methods like SPA and CMF cannot deploy on edge devices due to exceeding 10GB memory requirements (Table 1).

To mitigate the inefficiency of backpropagation, several recent studies have proposed lightweight alternatives via reducing the overhead of gradient-based updates. For example, MECTA (Hong et al., 2023) combines model pruning with entropy minimization to reduce gradient computation.

---
*Corresponding author.

EcoTTA (Song et al., 2023) replaces heavy convolutional blocks with lightweight meta-networks to lower backpropagation costs. Similarly, L-TTA (Shin & Kim, 2024) observes that shallow layers contribute most to adaptation and thus restricts updates to the stem layers, simplifying the process. More recently, some methods attempt to remove backpropagation altogether. FOA (Niu et al., 2024), for instance, performs derivative-free prompt search for Vision Transformers (ViTs) (Dosovitskiy et al., 2020), thereby eliminating backward passes and reducing memory usage. However, FOA still incurs high latency, as achieving competitive accuracy requires a large number of forward passes (e.g., 27).

A second major limitation of existing efficient TTA methods lies in their lack of architectural generality. While full backpropagation-based approaches are broadly applicable to both CNNs (He et al., 2016) and Transformers (Vaswani et al., 2017), most efficient variants are narrowly tailored. For instance, FOA is designed exclusively for ViTs via prompt tuning and cannot be applied to CNNs. Conversely, methods like EcoTTA and MECTA are tailored to ResNet-style CNNs that rely on batch normalization layers, rendering them ineffective for Transformer architectures.

In this paper, we introduce **PEA**, a **backpropagation-free** and **architecture-agnostic** method for efficient TTA. Our approach is motivated by a principled analysis of how domain shifts distort intermediate feature representations. Specifically, our analysis reveals that features from shifted domains consistently diverge from source-domain features through three structural transformations: (i) *mean shift*, which displaces global feature centroids, analogous to a **translation** of the distribution; (ii) *variance shift* that modifies the spread of features and inter-class spacing, corresponding to **scaling**, and (iii) *channel-wise covariance shift*, which modifies inter-feature correlations, effectively **rotating** the feature space and reorienting class relationships.

Grounded in these observations, PEA progressively aligns feature covariances at each model block during inference, thereby enhancing the quality of final-layer representations and improving prediction reliability. Specifically, PEA implements a two-forward-pass procedure: the first pass identifies the layer-wise shifts, and then, based on these shifts, assigns weights for each block to implement a covariance alignment across all layers' embeddings. Unlike prior methods, PEA is both backpropagation-free and architecture-agnostic, making it applicable to both CNNs and Transformers. This provides a unified and efficient solution to TTA. Our main contributions are as follows:

- Our analysis of intermediate embeddings uncovers the essence of domain shifts, which can be characterized as translations, scalings, and rotations of the embedding space.
- We propose PEA, an approach that adapts using only two forward passes per batch without backpropagation, allowing efficient adaptation with minimal memory and compute overhead.
- PEA is the first unified TTA framework that seamlessly generalizes across both CNNs and Transformers using identical procedures. Experiments on CIFAR-C and ImageNet-C demonstrate that it achieves competitive or superior performance compared to state-of-the-art methods, while maintaining high efficiency with successful deployment on resource-constrained edge devices.

## 2 RELATED WORK

**Conventional Test-Time Adaptation.** TTA has emerged as a practical solution for mitigating domain shifts that can severely degrade model reliability in deployment (Wang et al., 2024; Liang et al., 2025; Xiao & Snoek, 2024). The core idea is to update a pretrained model online using only the incoming unlabeled test batches, without requiring access to source data or ground-truth labels.

Early TTA studies primarily focused on updating the model's normalization layers. For example, simply recalibrating batch normalization (BN) statistics was found to recover some of the accuracy lost under distribution shifts (Benz et al., 2021). Building on this idea, entropy-based optimization techniques such as TENT (Wang et al., 2020) and EATA (Niu et al., 2022) update gradients online under the guidance of prediction entropy, often combined with sample filtering or dynamic reweighting to improve stability. These methods established the foundation for unsupervised TTA, which adapts models based solely on their confidence without relying on external labels.

In parallel, another branch of work leveraged the model's own predictions as supervision signals. These self-supervised strategies fine-tune the model with pseudo-labels generated from the current test batch. Representative examples include mean-teacher adaptation (Wang et al., 2022), meta-learned initialization for rapid convergence (Bartler et al., 2022), and improved label robustness via

symmetric cross-entropy (Döbler et al., 2023). More recent efforts further stabilized this process through ensembling (Marsden et al., 2024) and Kalman filter refinement (Lee & Chang, 2024).

Despite their differences, both unsupervised and self-supervised TTA methods share a key limitation: they rely on backpropagation during adaptation. The need to compute gradients and store intermediate activations largely increases memory and computation overhead, limiting their practicality on resource-constrained devices and motivating the development of more efficient alternatives.

**Efficient Test-Time Adaptation.** Recent TTA research has increasingly focused on improving efficiency from various angles. Memory-aware gradient-based methods aim to reduce the footprint of backpropagation. For example, T3A (Iwasawa & Matsuo, 2021) only adapt the final classifiers to make the adaptation lightweight, but suffer from the sub-optimal improvement. MECTA (Hong et al., 2023) prunes gradient paths and normalizes only selected layers to lower activation storage, while EcoTTA (Song et al., 2023) leverages compact meta-networks to minimize backpropagation overhead. L-TTA (Shin & Kim, 2024) enhances efficiency by restricting adaptation to shallow stem layers in CNNs, and TinyTTA (Jia et al., 2024) combines early-exit classifiers with ensembling for low-memory adaptation on microcontrollers.

Notably, forward-only approaches eliminate gradient computation entirely. For example, LAME (Boudiaf et al., 2022) adjusts classifier decision boundaries post hoc without any gradient updates, though its limited adaptability can reduce accuracy. FOA (Niu et al., 2024) employs derivative-free prompt optimization for Vision Transformers, substantially lowering memory usage but incurring high latency due to the large number of forward passes required.

Moreover, FOA includes an activation shifting module that is related to our work. However, while FOA applies a simple mean shift only to the final-layer CLS token and relies primarily on test-time prompt optimization with many forward passes, PEA is statistics-driven and architecture-agnostic: motivated by the empirical observation that domain shift induces systematic embedding changes across layers (mean, variance, and covariance) for both ViT and ResNet, PEA performs block-wise covariance alignment to progressively pull target embeddings back toward the source distribution. This produces a fine-grained layer-wise realignment of the representation.

Overall, existing efficient TTA methods either rely on backpropagation, leading to high memory and computational costs, or are limited to specific architectures (e.g., CNN-only or ViT-only designs). In contrast, our method presents a unified forward-only framework that delivers fast, memory-efficient adaptation while maintaining strong accuracy across both CNNs and Transformers.

## 3 MOTIVATION: ANALYSIS OF DOMAIN SHIFT

Although contemporary TTA methods have achieved empirical success, they often treat domain shift as a black-box problem, focusing on high-level strategies like entropy minimization and prompt tuning without exploring the root cause of performance degradation. This motivates our central question: **what is the essence of domain shift**? We approach this question from the perspective of the **embedding space**, hypothesizing that *misalignment in intermediate representations is a key driver of performance drop under domain shift*. To test this, we conducted an empirical analysis using a ViT model trained on CIFAR10 (source) and evaluated on CIFAR10-C with `Fog` (target). We applied t-SNE to visualize the intermediate embeddings from ViT block 3, focusing on three representative classes for the sake of clarity in illustration. The resulting visualizations consistently reveal three distinct structural transformations in the embedding space, as shown in Figure 1. We conduct more similar experiments and observe the same phenomenon, which can be found in Appendix A.

Our analysis reveals that, despite its varied forms, domain shift primarily manifests through three characteristic geometric changes in the embedding space:

**(i) Translation (Mean Shift).** As shown in Figure 1(a), the most fundamental effect of domain shift is a translation of the feature distribution. The global centroid of the target domain's embeddings is displaced relative to that of the source domain. As a result, the embedding magnitudes in the target shifted domain become misaligned with the parameters learned by the source model. While this is the most common form of shift addressed by conventional TTA methods, it is often only one part of a more complex problem.

**(ii) Scaling (Variance Shift).** Beyond a simple translation, domain shift significantly alters the scaling of the entire feature distribution, corresponding to a variance shift. As depicted in Figure

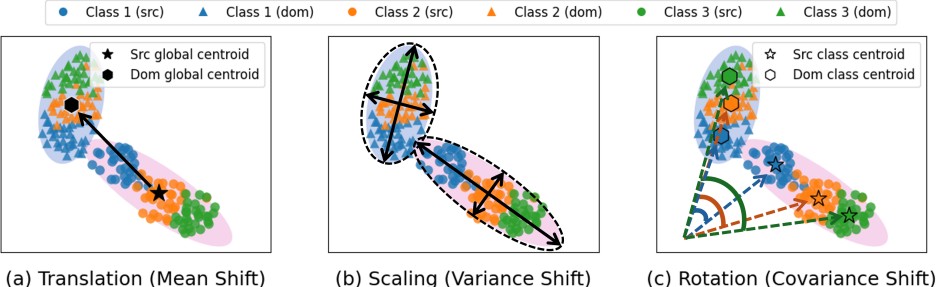

Figure 1: Impact of domain shift on intermediate layer embeddings. Feature distributions of three classes from block 3 of the ViT model are visualized. Each subfigure illustrates a different type of shift: translation, scaling, and rotation. More experiments can be found in Appendix A.

1(b), the global "cloud" of features changes its overall shape and density. Some layers may exhibit a more compact feature distribution, where the embeddings are compressed closer to their mean, while others become more dispersed, expanding outwards. This observation is consistent with the insight in GALA (Sahoo et al., 2025) and PALM (Maharana et al., 2025). This non-uniform scaling across layers cannot be corrected by simple global normalization; instead, it requires a layer-specific approach to align the variance change in feature space.

**(iii) Rotation (Channel-wise Covariance Shift).** Our most crucial observation is the presence of a covariance shift in the feature space. This indicates a systematic change in the correlation among the embedding dimensions. The shift mainly appears as a coherent geometric transformation of the feature cloud, resembling rotation and shearing. As visualized in Figure 1(c), this distortion goes beyond simple translation and scaling, fundamentally altering the relative orientation and arrangement of class clusters.

## 4 PROGRESSIVE EMBEDDING ALIGNMENT

Based on the analysis above, a natural TTA solution is to progressively realign the shifted embeddings toward the source distribution across model layers. However, applying such alignment presents two key challenges: (1) Since intermediate features are automatically learned and propagated through the model layers, even small misalignments at early layers can accumulate and cause significant degradation in deeper representations. (2) TTA typically operates with small batch sizes (e.g., 64 or fewer) on the devices, making it difficult to reliably estimate feature statistics.

To address these issues, we propose Progressive Embedding Alignment (PEA), a simple yet effective method that incrementally refines intermediate representations through robust covariance alignment. To tackle the challenge of accumulating errors, our method employs a **distance-aware weighted covariance alignment** strategy that progressively interpolates between the original and aligned embeddings based on their degree of shift, ensuring robustness and preventing over-correction. To overcome the challenges of small batch sizes, we introduce two techniques: an **exponential moving average (EMA)** to accumulate historical estimates of statistics, and lightweight **data augmentation** to diversify input samples and enrich the feature distribution observed at test time. Unlike many prior TTA methods that require updating model parameters to fit the shifted domain, *PEA is entirely backpropagation-free and architecture-agnostic, operating solely on intermediate features.* The complete PEA pipeline is summarized in Algorithm 1 (Appendix B).

### 4.1 DISTANCE-AWARE WEIGHTED COVARIANCE ALIGNMENT

The key objective of our method is to progressively realign the test-time intermediate features with the source-domain distribution at each block of the DNN. We achieve this using a Whitening-Coloring Transform (WCT) (Cho et al., 2019) that geometrically transforms the target-domain features to match the structure of the source domain. However, as we mentioned in the first challenge above, applying covariance alignment too aggressively risks over-correction and misalignment. To balance this, we introduce *a distance-aware weighting mechanism* that adaptively combines the original and aligned features based on their layer-specific statistical discrepancy. Our method operates

in two stages: an **offline stage** that extracts source statistics prior to deployment, and an **online stage** that performs dynamic alignment at test time through a two-forward-pass procedure.

**Offline Stage.** Prior to test-time deployment, we compute and store the source feature statistics for each block $l$ of the model using the training set. These include the source mean vector $\boldsymbol{\mu}_{s,l}$ and covariance matrix $\boldsymbol{\Sigma}_{s,l}$. These pre-computed statistics serve as the source geometry toward which we realign the test-time features. This offline process requires only a forward pass through the training data and does not involve any gradient computation or backpropagation. Once computed, the statistics require only minimal storage (about 30MB for ViT-Base) and enable deployment without ongoing source data access, making our approach practical for real-world deployment scenarios.

**Online Stage.** At test time, each incoming batch undergoes two forward passes. The first pass estimates the degree of domain shift at each layer to determine the appropriate alignment strength. The second pass then performs the actual feature alignment using WCT. Unlike prior forward-only methods (Niu et al., 2024) that require multiple runs to optimize prompts, our approach achieves adaptation with just *two forward passes*.

*Pass 1: Estimating Alignment Weights.* The goal of the first pass is to measure how much the current batch deviates from the source distribution at each block. To achieve this, we forward the test batch through the network to extract the intermediate feature activations $\boldsymbol{F}_l \in \mathbb{R}^{B \times N \times D}$. For each block $l$, we compute the batch mean $\boldsymbol{\mu}_{b,l}$ and variance $\boldsymbol{\sigma}_{b,l}^2$. These statistics characterize the current batch's distribution. To quantify the shift, we calculate a statistical distance between the batch and source distribution:

$$d_l = \|\boldsymbol{\mu}_{s,l} - \boldsymbol{\mu}_{b,l}\|_2 + \|\boldsymbol{\sigma}_{s,l}^2 - \boldsymbol{\sigma}_{b,l}^2\|_2 \tag{1}$$

This distance captures both center shift (translation) and scale mismatch at each layer. We then normalize these raw distances across all layers using min-max scaling to obtain the alignment weight $w_l \in [0, 1]$:

$$w_l = \frac{d_l - \min_l d_l}{\max_l d_l - \min_l d_l} \tag{2}$$

The weight $w_l$ reflects how strongly the features at block $l$ should be aligned: *layers with minimal shift receive near-zero weights (i.e., skip alignment), while those with high discrepancy are corrected more aggressively.*

*Pass 2: Performing Weighted Feature Alignment.* In the second forward pass, we reprocess the batch through the model and apply WCT-based alignment at each block. Let the updated test-time batch statistics be $\boldsymbol{\mu}_{t,l}$ and $\boldsymbol{\Sigma}_{t,l}$, which may be computed either from the current batch or from EMA tracking (see Section 4.2). We then apply the whitening-coloring transformation:

$$\boldsymbol{Y}_l = (\boldsymbol{F}_l - \boldsymbol{\mu}_{t,l})\boldsymbol{\Sigma}_{t,l}^{-1/2}\boldsymbol{\Sigma}_{s,l}^{1/2} + \boldsymbol{\mu}_{s,l} \tag{3}$$

In Eq. 3, we first whiten the test features by removing domain-specific variations using the target-domain mean and the square root of its covariance matrix. We then re-color the features with the source-domain covariance and mean to restore the geometry of the source distribution.

Instead of directly replacing the original feature with the aligned output, we blend them using the previously computed weight:

$$\boldsymbol{F}_l' = (1 - w_l)\boldsymbol{F}_l + w_l\boldsymbol{Y}_l \tag{4}$$

The combination of $\boldsymbol{F}_l$ and $\boldsymbol{Y}_l$ ensures that features are only shifted when necessary, maintaining stability for well-aligned layers while correcting mismatched ones.

One of the main computational bottlenecks in our alignment lies in the operations on covariance matrices, especially the computation of the matrix square root $\boldsymbol{\Sigma}^{1/2}$ and its inverse $\boldsymbol{\Sigma}^{-1/2}$. To perform this efficiently and stably, we use eigendecomposition tailored for symmetric positive semi-definite (SPSD) matrices. Given a covariance matrix $\boldsymbol{\Sigma}$, we first compute the eigendecomposition $\boldsymbol{\Sigma} = \boldsymbol{V}\boldsymbol{\Lambda}\boldsymbol{V}^\top$, where $\boldsymbol{V}$ contains the eigenvectors and $\boldsymbol{\Lambda}$ contains the eigenvalues. The square root and inverse square root are then computed as:

$$\boldsymbol{\Sigma}^{1/2} = \boldsymbol{V}\boldsymbol{\Lambda}^{1/2}\boldsymbol{V}^\top, \quad \boldsymbol{\Sigma}^{-1/2} = \boldsymbol{V}\boldsymbol{\Lambda}^{-1/2}\boldsymbol{V}^\top \tag{5}$$

This eigendecomposition simplifies the computation of the matrix square root and its inverse, effectively avoiding the high computational burden of general matrix operations. Overall, our method introduces minimal overhead: the eigendecomposition used for alignment is computationally efficient

due to the moderate feature dimensionality at each layer (typically 128 - 1024), and it is only applied during the forward pass. Crucially, our approach is entirely *gradient-free* and *model-agnostic* — it does not require backpropagation and task-specific tuning. All operations are performed on intermediate feature activations, allowing for seamless integration with a wide range of architectures (e.g., CNNs and ViTs) and low-latency deployment on resource-constrained devices.

## 4.2 ROBUST STATISTICS ESTIMATION VIA EMA

The effectiveness of the embedding alignment critically depends on the accurate estimation of the target domain statistics $(\boldsymbol{\mu}_{t,l}, \boldsymbol{\Sigma}_{t,l})$. However, test-time deployment, especially on resource-constrained devices equipped with limited memory, often necessitates small batch sizes (e.g., 64 or fewer), resulting in unreliable statistical estimates when derived from a single batch. To mitigate this issue, we maintain an Exponential Moving Average (EMA) strategy of the target feature statistics to accumulate historical batches to yield a more stable and robust estimation over time. For each new batch $i$, the EMA is updated with a momentum parameter $m$:

$$\boldsymbol{\mu}_{t,l}^{(i)} = (1-m)\,\boldsymbol{\mu}_{t,l}^{(i-1)} + m\,\boldsymbol{\mu}_{b,l}, \quad \boldsymbol{\Sigma}_{t,l}^{(i)} = (1-m)\,\boldsymbol{\Sigma}_{t,l}^{(i-1)} + m\,\boldsymbol{\Sigma}_{b,l} \quad (6)$$

While EMA ensures stability, it can be slow to adapt to *sudden and fast* domain shifts, causing the model to be anchored to outdated statistics. To solve this problem, we incorporate a **spike domain shift detection mechanism based on prediction entropy**.

Spike detection uses the model's prediction confidence as a signal for detecting a domain shift. A sudden drop in confidence (i.e., a sharp rise in entropy) often indicates that the model is encountering a new, unfamiliar domain (Ma et al., 2025). We track an EMA of the batch average prediction entropy, $E_{\text{ema}}$, and compare it to the instantaneous entropy of the current batch, $H_t$. A spike is flagged if the current entropy surpasses the historical average by a fixed threshold $\theta_{\text{ent}}$:

$$\text{Spike if: } H_t > E_{\text{ema}} + \theta_{\text{ent}} \quad (7)$$

If an entropy spike is detected, the EMA statistics $(\boldsymbol{\mu}_{t,l}, \boldsymbol{\Sigma}_{t,l})$ are immediately reset to those of the current batch. The detection module allows the model to rapidly adapt to the new data distribution, ensuring both stability during gradual shifts and agility during abrupt ones. The EMA update is computationally lightweight, involving only simple averaging per layer with negligible cost. Memory usage is also minimal, requiring storage of just two small tensors per block.

## 4.3 DATA ENRICHMENT VIA LIGHTWEIGHT AUGMENTATION

To further enhance the estimation of the target batch distribution, we introduce a lightweight data enrichment strategy based on *simple and low-cost augmentations* (Simonyan & Zisserman, 2014). These augmentations include common geometric transformations such as horizontal flips, random crops, and mild rotations. They are computationally inexpensive and preserve the semantic consistency of the domain. For each input image, we generate $K$ augmented views. This data augmentation is integrated into both forward passes during the online adaptation stage:

*Pass 1:* As described in Section 4.1, the first forward pass is used to estimate the layer-wise distribution discrepancy by computing the feature statistics of the current batch. To enhance the estimation under small batch sizes, we apply augmentation to each image and process the resulting $K$-view batch in the first forward pass. We then compute the alignment distance in Eq. 1 using this enriched batch, which results in more robust and stable weight estimation for each layer.

*Pass 2:* The second forward pass performs the actual alignment using the WCT transformation shown in Eq. 3. As in the first pass, we augment the batch into $K$ views and apply the WCT alignment across all views. After obtaining $K$ sets of aligned predictions, we aggregate them through uniform averaging:

$$\mathbf{pred}_{\text{final}} = \frac{1}{K} \sum_{k=1}^{K} \mathbf{logits}_k \quad (8)$$

The feature enrichment and ensembling not only improve the stability of embedding alignment but also enhance final predictions by incorporating multiple complementary views of the data. Despite introducing multiple views per input, the augmentations are lightweight and require no additional model parameters or backward passes. As a result, the added cost is limited to repeated forward

Table 1: Comparison of accuracy (%) on ImageNet-C using ViT-Base and ResNet-50 with memory consumption on server. Aug and BP indicate whether the approaches utilize data augmentation and backpropagation. In FOA, F specifies how many forward passes per batch.

| Model | Aug | BP | Methods | gauss. | shot | impul. | defoc. | glass | motion | zoom | snow | frost | fog | brigh. | contr. | elast. | pixel. | jpeg | Avg. | Mem. (MB) | Latency (s/batch) |
|---|---|---|---|---|---|---|---|---|---|---|---|---|---|---|---|---|---|---|---|---|---|
| ViT | ✗ | ✗ | No Adapt | 56.7 | 56.8 | 57.5 | 46.9 | 35.6 | 53.1 | 44.8 | 62.2 | 62.5 | 65.7 | 77.6 | 32.6 | 46.0 | 66.9 | 67.6 | 55.5 | 858 | 0.18 |
| | ✗ | ✓ | SAR | 59.9 | 62.2 | 62.8 | 54.1 | 54.1 | 59.1 | 54.5 | 63.5 | 65.6 | 65.3 | 78.2 | 64.4 | 58.3 | 69.2 | 69.7 | 62.7 | 6181 | 0.59 |
| | ✗ | ✓ | Tent | 57.1 | 58.1 | 59.2 | 44.7 | 43.2 | 56.6 | 50.6 | 62.8 | 60.5 | 65.2 | 78.0 | 59.7 | 49.8 | 68.2 | 68.6 | 58.8 | 6108 | **0.31** |
| | ✗ | ✓ | EATA | 57.3 | 59.1 | 59.9 | 53.6 | 49.4 | 58.2 | 51.8 | 63.0 | 62.9 | 65.7 | 77.8 | 62.0 | 55.7 | 65.8 | 68.7 | 60.7 | 6108 | **0.31** |
| | ✗ | ✗ | FOA (F = 27) | 61.5 | 63.5 | 64.3 | 56.9 | 55.1 | 61.0 | 60.9 | 68.4 | 70.9 | 73.6 | 80.9 | 66.0 | 61.8 | 73.5 | 73.6 | 66.1 | **870** | 3.33 |
| | ✗ | ✗ | FOA (F = 9) | 60.5 | 63.1 | 63.9 | 54.6 | 48.5 | 60.4 | 57.2 | 66.8 | 69.6 | 71.5 | 80.9 | 66.6 | 55.9 | 72.9 | 72.8 | 64.3 | **870** | 1.25 |
| | ✓ | ✓ | CMF | 60.0 | 61.2 | 60.9 | 56.6 | 56.8 | 62.4 | 60.8 | 69.3 | 67.9 | 72.7 | 78.8 | 65.2 | 69.4 | 73.8 | 72.0 | 65.9 | 10404 | 0.53 |
| | ✓ | ✓ | SPA | 61.7 | 64.0 | 63.0 | 50.7 | 58.3 | 63.0 | 59.1 | 68.2 | 65.5 | 67.9 | 77.7 | 63.9 | 67.3 | 72.6 | 66.0 | 64.6 | 10902 | 0.50 |
| | ✗ | ✗ | PEA | 57.7 | 58.4 | 58.9 | 53.2 | 50.4 | 59.8 | 60.4 | 69.1 | 68.7 | 72.9 | 80.2 | 63.5 | 69.9 | 72.0 | 72.0 | 64.5±0.0 | 887 | **0.31±0.1** |
| | ✓ | ✗ | PEA + Aug | 61.2 | 61.5 | 62.1 | 55.6 | 52.4 | 61.4 | 62.1 | 70.4 | 70.7 | 74.5 | 80.9 | 65.8 | 71.7 | 73.4 | 73.6 | **66.5±0.1** | 1867 | 0.59±0.2 |
| ResNet | ✗ | ✗ | No Adapt | 22.2 | 23.7 | 21.3 | 20.0 | 10.2 | 21.6 | 26.1 | 31.6 | 33.1 | 39.3 | 67.7 | 25.4 | 14.0 | 13.1 | 47.3 | 27.8 | 817 | 0.17 |
| | ✗ | ✓ | Tent | 14.4 | 17.9 | 14.2 | 14.3 | 15.3 | 27.8 | 51.1 | 41.9 | 43.6 | 59.8 | 69.4 | 27.4 | 45.3 | 44.2 | 46.4 | 35.5 | 5901 | 0.36 |
| | ✗ | ✓ | EATA | 15.4 | 20.6 | 18.6 | 17.4 | 19.7 | 32.5 | 44.9 | 44.5 | 47.4 | 60.8 | 70.0 | 34.4 | 49.0 | 51.0 | 50.9 | 38.5 | 5965 | 0.36 |
| | ✗ | ✓ | MECTA | 19.3 | 22.9 | 18.6 | 16.1 | 18.0 | 31.6 | 45.0 | 44.9 | 45.1 | 63.0 | 71.1 | 33.5 | 47.7 | 53.2 | 39.4 | 38.0 | 4425 | 0.50 |
| | ✗ | ✓ | EcoTTA | 3.9 | 6.2 | 3.5 | 7.5 | 9.8 | 24.2 | 41.8 | 43.3 | 31.9 | 60.7 | 68.8 | 16.5 | 47.1 | 46.2 | 46.2 | 30.5 | 5177 | 0.63 |
| | ✗ | ✓ | L-TTA | 16.8 | 24.0 | 22.6 | 12.1 | 16.7 | 23.3 | 33.6 | 42.0 | 44.8 | 57.3 | 66.7 | 16.6 | 43.5 | 49.4 | 47.8 | 34.5 | 3373 | **0.25** |
| | ✓ | ✓ | CMF | 35.0 | 35.1 | 36.6 | 19.2 | 27.5 | 34.5 | 42.9 | 47.9 | 47.6 | 60.2 | 69.7 | 38.2 | 51.0 | 54.7 | 55.7 | 43.7 | 10413 | 0.38 |
| | ✓ | ✓ | LAW | 19.6 | 26.2 | 28.8 | 22.5 | 27.1 | 36.1 | 45.9 | 42.1 | 40.2 | 53.7 | 64.5 | 32.8 | 52.5 | 55.1 | 52.8 | 39.9 | 11734 | 0.70 |
| | ✗ | ✗ | PEA | 22.6 | 25.0 | 22.0 | 26.0 | 23.1 | 37.3 | 48.2 | 49.7 | 48.7 | 64.8 | 73.5 | 51.1 | 53.0 | 43.3 | 52.2 | 42.7±0.1 | **983** | 0.36±0.2 |
| | ✓ | ✗ | PEA + Aug | 26.2 | 28.3 | 25.8 | 27.5 | 24.8 | 38.8 | 49.6 | 51.3 | 50.4 | 65.6 | 74.0 | 52.4 | 55.1 | 47.9 | 54.1 | **44.8±0.2** | 2397 | 0.56±0.2 |

passes with minor geometric transforms, making this approach highly efficient and practical even on memory-constrained edge devices.

**Fundamental Methodological Difference of PEA:** Existing TTA methods typically update the affine parameters of normalization layers through backpropagation, i.e., they *adapt the model to fit the shifted domains*, using techniques such as entropy minimization and data augmentation. However, as discussed in (Press et al., 2024), the absence of ground-truth labels at test time often cause embedding drifts over successive iterations, resulting in suboptimal performance or even leading to catastrophic forgetting.

In contrast, our approach adopts a fundamentally different strategy: rather than modifying the model, we *align the shifted embeddings with the source distribution*. This eliminates the need for *backpropagation*, ensuring that the original model parameters remain intact and robust, thereby completely mitigating catastrophic forgetting.

## 5 EXPERIMENTS

### 5.1 DATASETS AND BASELINES

**Datasets and Models.** Following recent works in TTA (Shin & Kim, 2024; Niu et al., 2024), we conduct a comprehensive evaluation across multiple datasets. Specifically, we use CIFAR10-C, CIFAR100-C, and ImageNet-C, each of which introduces 15 common corruption types applied to the original test sets. We adopt the most severe corruption level (severity = 5) and batch size of 64 throughout all experiments. To simulate a realistic online domain shift scenario, we follow the **lifelong continual** test-time adaptation setting in CoTTA (Wang et al., 2022; Niu et al., 2022), where corrupted samples are streamed sequentially at test time. Compared to always adapting each domain from the source domain, our continual setting is more realistic and challenging.

For backbone models, we adopt both ResNet-50 (He et al., 2016) and ViT-Base (Dosovitskiy et al., 2020) on the ImageNet-C and CIFAR100-C datasets. For CIFAR10-C, we evaluate using ResNet-50 and ViT-Tiny to account for the dataset's smaller scale. This diverse selection demonstrates that our method generalizes effectively across both CNN and Transformer-based architectures.

**Baselines.** We compare our proposed PEA with several efficient TTA approaches as well as state-of-the-art performance-driven methods. For efficient CNN-based TTA, we include EcoTTA (Song et al., 2023), MECTA (Hong et al., 2023), and L-TTA (Shin & Kim, 2024). For ViT-specific adaptation, we evaluate FOA (Niu et al., 2024), which performs forward-only prompt optimization. We also evaluate entropy minimization-based methods including Tent (Wang et al., 2020), EATA (Niu et al., 2022), and SAR (Niu et al., 2023). Finally, we include recent state-of-the-art approaches based on pseudo-labeling and data augmentation: CMF (Lee & Chang, 2024), LAW (Park et al., 2024) and SPA (Niu et al., 2025)[1]. Details of the implementation and additional clarifications are provided in Appendix C.

---

[1]The SPA results reported in the original paper are obtained under a single-domain adaptation setting, where the model is reset before each corruption. In our experiments, we use a more challenging lifelong TTA setting, where the model adapts continuously across all domains without reset.

Table 2: Adaptation accuracy (%) on CIFAR10-C and CIFAR100-C using ViT and ResNet.

| Model | Dataset | Method | | | | | | | | | |
|---|---|---|---|---|---|---|---|---|---|---|---|
| | | No Adapt | SAR | Tent | EATA | FOA (F=9) | FOA (F=27) | CMF | SPA | PEA | PEA with Aug |
| ViT | CIFAR10-C | 61.6 | 61.3 | 61.2 | 61.6 | 67.2 | 68.0 | 73.0 | 71.7 | 75.7±0.2 | **77.0±0.2** |
| | CIFAR100-C | 76.5 | 75.8 | 76.2 | 76.5 | 81.9 | 83.3 | 83.3 | 76.2 | 83.7 | **84.7±0.1** |
| | Dataset | No Adapt | TENT | EATA | MECTA | EcoTTA | L-TTA | CMF | LAW | PEA | PEA with Aug |
| ResNet | CIFAR10-C | 62.5 | 81.2 | 81.3 | 82.0 | 80.2 | 81.2 | 78.6 | 81.8 | 81.8±0.1 | **83.4±0.1** |
| | CIFAR100-C | 33.5 | 49.2 | 50.0 | 49.7 | 45.6 | 50.9 | 48.8 | 51.0 | 53.9±0.2 | **54.6±0.3** |

## 5.2 MAIN RESULTS ON IMAGENET-C

Table 1 presents the classification accuracy and variation (averaged over 5 runs with different random seeds) for each domain, together with the memory consumption and per-batch inference latency measured on the server.

For ViT-Base, without adaptation, the baseline ViT model achieves an average accuracy of 55.5%. Although existing methods such as Tent (Wang et al., 2020) and EATA (Niu et al., 2022) offer moderate improvements to 58.8% and 60.7%, they incur substantial memory overhead (more than 6 GB) due to backpropagation-based updates. More recent backprop-free method FOA (Niu et al., 2024) and SOTA SPA (Niu et al., 2025) achieve stronger accuracy (up to 66.1% and 64.6%) but with high latency (up to 3.33s) or memory consumption (over 10 GB). By contrast, our PEA achieves 64.5% accuracy with only 887MB of memory and 0.31s latency. When combined with augmentation (PEA + Aug), performance further improves to 66.5%, surpassing FOA with better latency. This demonstrates that PEA not only provides competitive accuracy but also delivers exceptional memory and latency efficiency, making it highly suitable for real-time or on-device deployment.

For ResNet-50, TTA baselines such as Tent, EATA, and CMF improve performance to up to 43%, but again at the cost of large memory (more than 5.9 GB) and higher compute demand. PEA outperforms all low-cost adaptation methods with an average accuracy of 42.7%, using only 983MB of memory. With augmentation, it reaches 44.8%, outperforming all existing backprop-free methods like EcoTTA and L-TTA by a large margin.

*Efficiency and Accuracy Trade-off:* Our approach achieves a highly favorable balance between robustness and efficiency. Unlike backpropagation-based TTA methods, PEA delivers strong adaptation performance while consuming significantly less memory and maintaining low latency. This lightweight yet effective design makes PEA highly suitable for practical deployment, especially in resource-constrained devices or real-time systems, we will further discuss it in Section 5.6.

## 5.3 RESULTS ON CIFAR10-C AND CIFAR100-C

We also evaluate the performance of PEA on the CIFAR10-C and CIFAR100-C using both ViT and ResNet. As shown in Table 2 (for more details see Section D.1), PEA consistently outperforms existing TTA approaches. Notably, under the ViT backbone, PEA achieves 77.0% accuracy on CIFAR10-C and 84.7% on CIFAR100-C when lightweight augmentation is applied, substantially outperforming augmentation-based baselines like CMF and SPA. Even without augmentation, PEA attains competitive results (75.7% and 83.7%), demonstrating its intrinsic robustness. Similar trends are observed with the ResNet backbone, where PEA achieves 83.4% on CIFAR10-C and 54.6% on CIFAR100-C, again outperforming strong baselines including MECTA, EcoTTA, and L-TTA.

In addition, we observe that augmentation-based methods such as CMF and SPA show relatively limited gains on these small-scale datasets compared to their performance on larger dataset ImageNet-C. This suggests that excessive reliance on augmentation alone may not generalize well across dataset scales. In contrast, PEA demonstrates strong generalization across both model architectures and dataset types. Importantly, it achieves this *without updating any model parameters* and is entirely *backprop-free*, making it naturally compatible with both CNN and Transformer architectures.

## 5.4 RESULTS ON SMALL BATCH SIZE

Table 3 presents the performance of our method under varying batch sizes (BS = 4, 16, 64) on both CIFAR100-C and ImageNet-C, using ResNet-50 and ViT-Base. We observe that while accuracy slightly drops as the batch size decreases, our method retains high performance even under very small batches. On CIFAR100-C, the ViT-Base model achieves 77.0% with BS=64 and maintains a strong 70.0% even with BS=4, a modest 7.0% drop. In contrast, the ResNet-50 model sees a smaller absolute decline (from 54.6% to 51.8%), but its overall accuracy remains much lower. A similar

Table 3: Results on small batch sizes on CIFAR100-C and ImageNet-C.

| Datasets | Model | Accuracy (%) | | |
|---|---|---|---|---|
| | | BS = 4 | BS = 16 | BS = 64 |
| CIFAR100-C | ResNet-50 | 51.8 | 54.0 | 54.6 |
| | ViT-Base | 70.0 | 75.7 | 77.0 |
| ImageNet-C | ResNet-50 | 41.7 | 44.0 | 44.8 |
| | ViT-Base | 63.3 | 65.8 | 66.5 |

Table 4: Evaluation on Jetson Orin Nano using CIFAR100-C with batch size of 64. Methods marked as incompatible (✗) fail due to insufficient memory on the target device (3.5 GB). Memory requirements for them are shown in Table 1.

| ViT-Base | | | ResNet-50 | | |
|---|---|---|---|---|---|
| Method | Latency (s/batch) | Memory (MB) | Method | Latency (s/batch) | Memory (MB) |
| No Adapt | 3.5 | 901 | No Adapt | 0.9 | 810 |
| SAR | ✗ | ✗ | Tent | ✗ | ✗ |
| Tent | ✗ | ✗ | EATA | ✗ | ✗ |
| EATA | ✗ | ✗ | MECTA | ✗ | ✗ |
| FOA (F = 9) | 98.9 | 920 | EcoTTA | ✗ | ✗ |
| CMF | ✗ | ✗ | L-TTA | 1.4 | 3249 |
| SPA | ✗ | ✗ | CMF | ✗ | ✗ |
| PEA | **4.1±0.2** | **1011** | PEA | **3.0±0.1** | **976** |
| PEA + Aug | 9.8±0.3 | 2322 | PEA + Aug | 7.2±0.2 | 2388 |

Table 5: Adaptation accuracy (%) on mixed domain on CIFAR100-C dataset.

| ViT-Base | No Adapt | Tent | EATA | SAR | FOA (K=9) | CMF | SPA | PEA |
|---|---|---|---|---|---|---|---|---|
| | 61.6 | 61.2 | 61.2 | 62.0 | 62.6 | 69.0 | 71.4 | 72.0 |

| ResNet-50 | No Adapt | Tent | EATA | MECTA | EcoTTA | CMF | L-TTA | PEA |
|---|---|---|---|---|---|---|---|---|
| | 33.5 | 17.4 | 16.5 | 40.2 | 7.3 | 7.3 | 13.4 | 47.4 |

trend is observed on ImageNet-C, where ViT-Base drops by 3.2% and ResNet by 3.1%. As shown in Table 9 in Appendix D.3, our method performs better than other baselines. Further, we evaluate our method under extremely small batch sizes (BS=1 and BS=2) to simulate a streaming inference setup. The results are reported in Section D.4.

## 5.5 RESULTS ON MIXED DOMAINS SETTING

We further evaluate PEA under a *mixed-domain* setting on CIFAR100-C, where all 15 corruption types (severity 5) are merged into a single pool and randomly shuffled. Consequently, each mini-batch contains samples from multiple corruptions (on average $64/15 \approx 4.3$ samples per corruption), emulating a realistic deployment scenario with rapid and unpredictable domain shifts *within* a batch. This setting is substantially more challenging than the single-domain protocol.

Table 5 summarizes the mixed-domain results. With ViT-Base, PEA achieves 72.0% accuracy, improving over the non-adapted source model by 10.4% and outperforming all baselines. Notably, entropy-minimization methods (Tent and EATA) provide no gains (61.2%), indicating that simply updating on heterogeneous batches can be ineffective, while even strong baselines such as CMF (71.4%) remain below PEA. On the more challenging ResNet-50 backbone, the gap is larger: Tent and EATA substantially degrade performance (17.4% and 16.5%) and several methods struggle (e.g., EcoTTA 7.3% and L-TTA 13.4%), whereas PEA reaches 47.4% and surpasses the strongest baseline, MECTA which performs 40.2%, demonstrating robust adaptation under rapid and irregular domain shifts. To further highlight the effectiveness of PEA under mixed-domain shifts, we provide additional visualizations in Section F.

## 5.6 EVALUATION ON EDGE DEVICE

To assess practical deployability, we evaluate the system performance on the Jetson Orin Nano, a resource-constrained edge device with 8 GB of shared memory, only 3.5 GB of which is accessible to deep learning applications due to OS and system overhead. We test all methods under a default setting with batch size 64 on CIFAR100-C. Table 4 reports both the latency (in seconds per batch) and peak memory usage (in MB) for ViT-Base and ResNet-50 backbones.

Due to limited memory, many TTA methods fail to run on-device, especially those requiring back-propagation (e.g., Tent, EATA, MECTA, SAR). In contrast, our method (PEA) successfully runs on both backbones, maintaining reasonable latency (4.1s for ViT, 3.0s for ResNet) and modest memory usage (1011MB and 976MB, respectively). With augmentation enabled, performance trade-offs increase modestly, but still remain within edge constraints. While both FOA and L-TTA are compatible with edge devices, FOA incurs extremely high latency, rendering it impractical for real-time applications. In contrast, L-TTA is fast but consistently underperforms in accuracy across all three

datasets, as discussed in Section 5.2 and 5.3. Notably, the forward-only design of PEA ensures compatibility with edge settings, where low memory footprint and gradient-free inference are critical, showing its strong potential for real-world deployment without sacrificing adaptation effectiveness.

## 5.7 ABLATION STUDY

We conduct an ablation study to quantify the contribution of each major component in PEA using the ViT-Base model on both CIFAR100-C and ImageNet-C. Table 6 summarizes the incremental performance improvements by the proposed components. Starting from the unadapted baseline, introducing only the covariance alignment module (Cov Align Only) brings a significant gain on CIFAR100-C (from 61.6% to 67.0%), demonstrating that aligning feature second-order statistics is a strong and lightweight signal for domain correction. However, this setting results in a sharp drop on ImageNet-C (down to 25.2%), due to over-alignment across all layers. Since ImageNet is more challenging and features greater domain complexity, the per-batch estimation of the target distribution becomes less reliable, leading to misaligned feature transformations.

Table 6: Ablation study of PEA using ViT-Base model on CIFAR100-C and ImageNet-C.

| Ablation | Acc. (CIFAR100-C) | Acc. (ImageNet-C) |
|---|---|---|
| No Adapt | 61.6 | 55.5 |
| Cov Align Only | 67.0 | 25.2 |
| + Weighting | 68.3 | 52.9 |
| + Weighting, EMA | 75.7 | 64.5 |
| + Weighting, EMA, Aug | 77.0 | 66.5 |

Adding the layer-wise distance-based weighting mechanism (+ Weighting) mitigates the misalignment on ImageNet-C, boosting performance from 25.2% to 52.9%. This highlights the importance of selectively applying alignment only to blocks that exhibit significant distributional shifts. The improvement on CIFAR100-C is more modest but still positive, suggesting that the weighting scheme contributes to robustness across datasets. Incorporating exponential moving average (EMA) for estimating test-time statistics (+ Weighting, EMA) provides a large boost in both datasets (75.7% on CIFAR100-C and 64.5% on ImageNet-C). The EMA strategy accumulates stable statistics over time, which is especially beneficial when the test-time batch size is small or noisy. This component ensures the alignment is based on reliable statistics rather than volatile per-batch estimates. Finally, adding data enrichment via lightweight augmentations (+ Weighting, EMA, Aug) yields the highest accuracy 77.0% on CIFAR100-C and 66.5% on ImageNet-C. The multiple views not only help stabilize the estimation of target statistics but also improve final predictions via ensemble averaging.

Overall, each component contributes complementary benefits to the final performance, and their combination enables PEA to maintain high accuracy under diverse corruptions while being backprop-free and resource-efficient. See Appendix D.5 for more hyperparameter evaluations.

## 6 CONCLUSION

This work begins by revisiting the impact of domain shift on intermediate model embeddings, identifying three core transformations: mean shift (translation), variance shift (scaling), and channel-wise covariance shift (rotation), which systematically distort the feature space across layers. Motivated by this insight, we propose PEA, a lightweight, backpropagation-free, and architecture-agnostic test-time adaptation approach that progressively aligns embeddings through layer-wise covariance correction using only two forward passes. Experiments across 3 datasets, including evaluations on resource-constrained edge devices, demonstrate that PEA achieves state-of-the-art accuracy and efficiency, offering a practical and generalizable solution for robust real-world deployment.

*Limitations.* While PEA provides a lightweight, backpropagation-free solution that generalizes across model architectures, it requires extracting source statistics from the training data prior to deployment. While this is acceptable in the standard TTA setting (Song et al., 2023; Niu et al., 2024), such source statistics may not always be available in some practical scenarios. That said, PEA does not require access to the full source dataset: using only 10% of the training data to compute these statistics is sufficient to maintain strong performance (see Section D.5.2 for detailed results).

## 7 ETHICS STATEMENT

This work does not involve human subjects, sensitive personal data, or applications with direct societal or ethical risks. The datasets used in our experiments are publicly available benchmarks that have been widely adopted in the research community. We believe the contributions of this paper align with the ICLR Code of Ethics.

## 8 REPRODUCIBILITY STATEMENT

We have taken several steps to ensure the reproducibility of our results. All datasets employed in this study are publicly available and described in the main text. Detailed descriptions of model architectures, hyperparameters, and training protocols are provided in the paper and appendix. Furthermore, we provide pseudocode and implementation details in the appendix, and the complete source code will be made available if the paper is accepted.

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

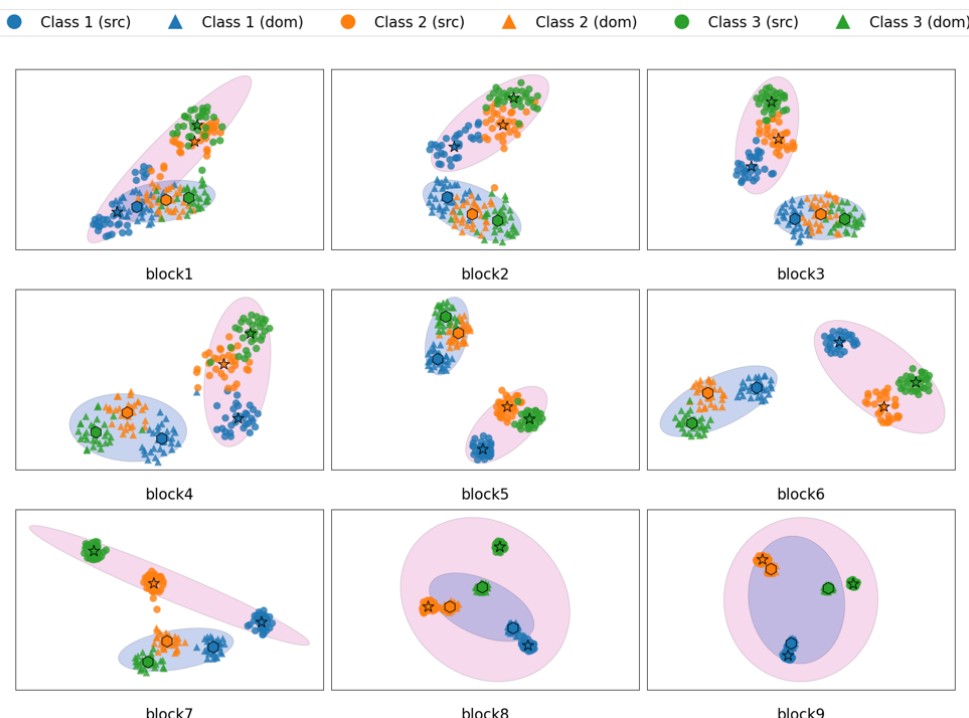

Figure 2: Impact of domain shift on intermediate layer embeddings using **CIFAR10-C**. We visualize the features of class plane, dog, and frog in Domain **Fog** across ViT block 1 to block 9.

## A    DOMAIN SHIFT IN THE EMBEDDING SPACE

As discussed in Section 3, domain shifts manifest as structural distortions in the intermediate feature space of deep models. These distortions—namely mean shift, variance shift, and covariance shift—occur consistently across all layers of the network. In this section, we provide additional empirical evidence to support this analysis by visualizing feature distributions under domain shift using ViT models on CIFAR10-C.

**Visualization of Feature Shift Across Layers and Domains.** We conduct a detailed visualization of intermediate features extracted from ViT blocks 1 through 9 on two corruption types from CIFAR10-C: Fog and Gaussian Noise. We focus on three representative classes—*plane*, *dog*, and *frog*—to illustrate how domain shift affects the geometry of class embeddings at different depths of the model.

Figures 2, 3 and 4 show the progressive deformation of class-wise embeddings under these two corruption domains. These results complement our earlier analysis and reveal consistent patterns across domains and layers.

From the visualizations, we observe that: 1) All layers are affected by geometric distortions. Across all blocks, we observe consistent evidence of (i) **mean shift**, where class centers drift from their source positions; (ii) **variance shift**, indicated by altered spread and scale of feature clusters; and (iii) **channel-wise covariance shift**, where the orientation and shape of the clusters change due to altered inter-channel relationships; 2) The severity of distortion varies across layers. Different layers exhibit different sensitivities to each type of transformation; 3) Different domains impact features in distinct ways. Although both Fog and Gaussian Noise induce all three types of shifts, the degree and pattern of deformation vary. This reflects the domain-specific characteristics of the corruption types—e.g., Fog tends to cause smoother global drifts, while Gaussian Noise leads to more irregular scatter.

Moreover, in domains such as Fog, which are relatively easy to adapt, the transformations at deeper layers are more faithful than at shallow layers, suggesting the architecture can progressively correct

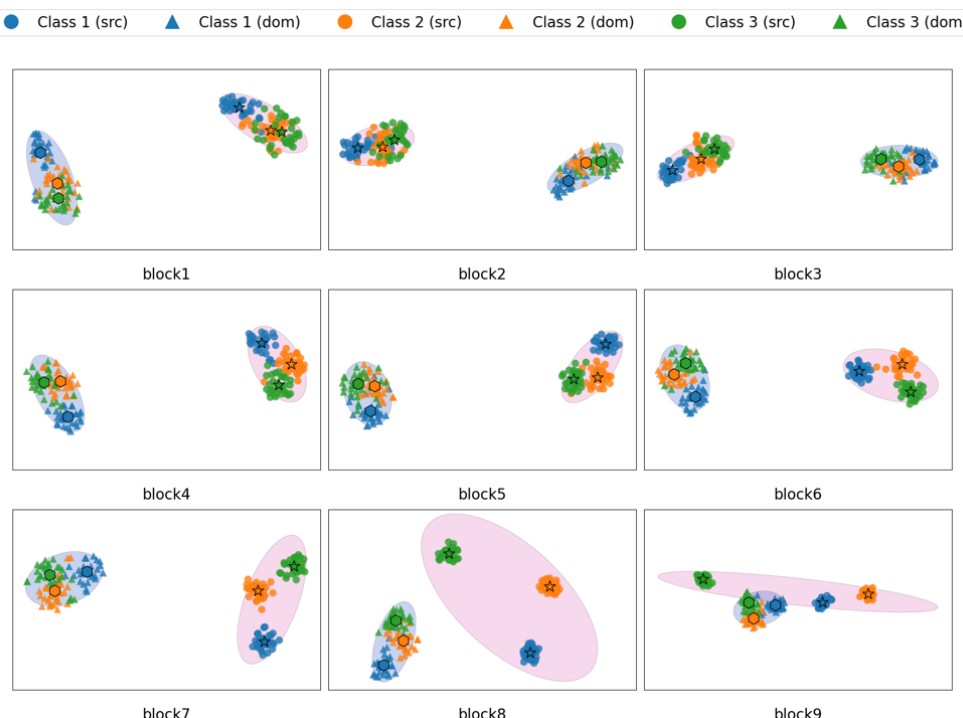

Figure 3: Impact of domain shift on intermediate layer embeddings using **CIFAR10-C**. We visualize the features of class plane, dog and frog in Domain **Gaussian Noise**.

the shift as features propagate. In contrast, in domains such as `Gaussian Noise`, deeper layer features deteriorate (pronounced scaling shrinkage and rotation), indicating that domain characteristics strongly shape the final representations and can hinder self-correction in depth.

These insights reinforce the central hypothesis of our work: *domain shift induces systematic, layer-wise geometric transformations in the embedding space.* They also motivate our proposed method, which explicitly corrects such distortions through progressive covariance alignment at each intermediate block.

**Visualization of Feature Shift on CIFAR100-C.** To further validate that the observed structural shifts in embedding space are not specific to CIFAR10-C, we extend our visualization to CIFAR100-C. We select three representative classes—pine tree, bicycle, and bee—from the CIFAR100-C dataset and examine their intermediate representations under domain shift caused by Shot Noise, a common corruption in CIFAR100-C. The results are shown in Figure 5.

Similar to the patterns observed in CIFAR10-C, we find that domain shift consistently induces systematic geometric transformations in the embedding space: mean shift (translation), variance shift (scaling) and covariance shift (rotation). While the inter-class topology is often preserved, these structural distortions displace features away from the decision boundaries, ultimately degrading classification performance. In particular, even though class relationships remain recognizable, the shifted features can no longer be correctly classified due to their increased distance from the source-aligned classification regions.

These shifts manifest across multiple blocks of the ViT model, reinforcing our claim that domain shift affects not only the output layer but also the intermediate representations in a systematic and structured manner. The consistency of these patterns across both CIFAR10-C and CIFAR100-C highlights the generality of our observation and motivates the need for intermediate-layer realignment strategies, such as the one introduced in PEA.

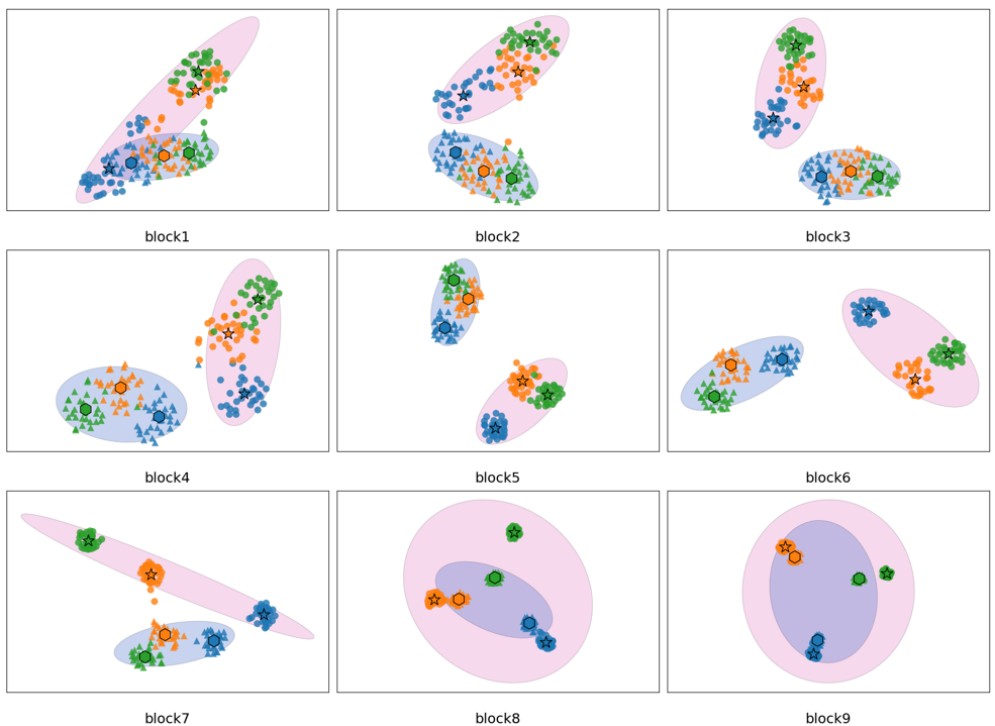

Figure 4: Impact of domain shift on intermediate layer embeddings using **CIFAR10-C**. We visualize the features of class plane, dog and frog in Domain **Defocus Blur**.

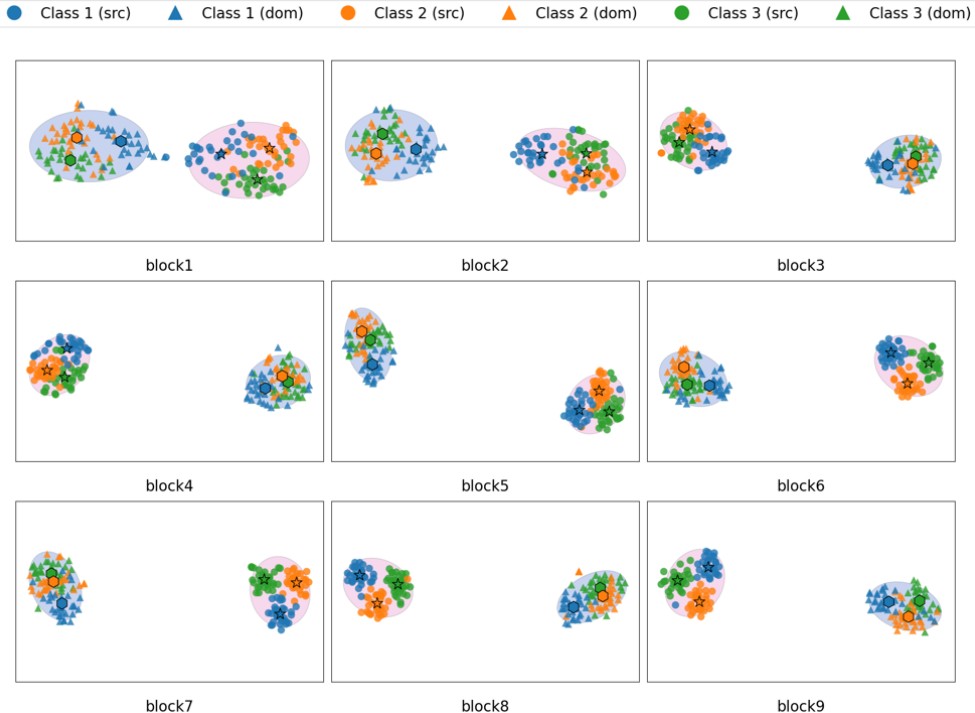

Figure 5: Impact of domain shift on intermediate layer embeddings using **CIFAR100-C**. We visualize the features of class pine tree, bicycle and bee in Domain **Shot Noise**.

---

**Algorithm 1** PROGRESSIVE EMBEDDING ALIGNMENT (PEA)

---

**Require:** source stats $\{\boldsymbol{\mu}_{s,l}, \boldsymbol{\Sigma}_{s,l}\}_{l=1}^{L}$, test batch $\boldsymbol{X}$, views $K$

1: **Pass 1: Estimate alignment weights**
2: Augment input: $\boldsymbol{X}_{\text{aug}} = \text{Augment}(\boldsymbol{X}, K)$
3: Extract block features $\boldsymbol{F}_l$; compute alignment weights $w_l$ using `Eq.1` and `Eq.2`
4: **Pass 2: Align and predict**
5: Extract $\boldsymbol{F}_l$ from $\boldsymbol{X}_{\text{aug}}$
6: **for** each block $l$ **do**
7:    Update $\boldsymbol{\mu}_{t,l}, \boldsymbol{\Sigma}_{t,l}$ with EMA (`Eq.6`)
8:    Apply covariance alignment: $\boldsymbol{Y}_l = (\boldsymbol{F}_l - \boldsymbol{\mu}_{t,l})\boldsymbol{\Sigma}_{t,l}^{-1/2}\boldsymbol{\Sigma}_{s,l}^{1/2} + \boldsymbol{\mu}_{s,l}$     // `Eq.3`
9:    Fuse: $\boldsymbol{F}'_l = (1 - w_l)\boldsymbol{F}_l + w_l\boldsymbol{Y}_l$     // `Eq.4`
10: **end for**
11: Compute logits for all views: $\textbf{logits}_k = \text{Classifier}(\boldsymbol{Z}_L)$
12: Final prediction: $\textbf{pred}_{\text{final}} = \frac{1}{K}\sum_{k=1}^{K}\textbf{logits}_k$     // `Eq.8`
13: **return** $\textbf{pred}_{\text{final}}$

---

## B    PSEUDO CODE

To clarify the workflow of our PEA, we present its pseudo-code in Algorithm 1, which outlines the step-by-step process of progressively aligning domain-shifted features toward the source distribution during test-time adaptation.

## C    IMPLEMENTATION DETAILS

We use a default batch size of 64 for all evaluations, consistent with prior works (Niu et al., 2022; Lee & Chang, 2024). All methods are implemented and tested on a server equipped with an NVIDIA A5000 Ada GPU. To evaluate the feasibility of deployment under real-world constraints, we also compare PEA with efficient TTA baselines on an edge device: the Jetson Orin Nano, which includes a Cortex-A78AE CPU and an 8GB shared RAM mobile GPU.

For our PEA, we set the EMA momentum to $m = 0.02$ to ensure a stable yet responsive estimate of feature statistics. To detect domain shifts, we use an entropy spike threshold $\theta_{\text{ent}} = 1.0$; when this is exceeded, the EMA statistics are reset. We tested EMA momentum values of 0.01, 0.02, 0.05, 0.1 and found the performance to be robust, with accuracy fluctuations of at most 1% (see D.5). For augmentation, we use random horizontal flips and random resized crops (scale = 0.9). Each input generates $K = 2$ augmented views, which, combined with the original, produce 3 views used for prediction ensembling. Since the augmentation is an optional technique based on the available memory, we also report the results without augmentation. All other hyperparameters, such as learning rates and optimization settings for baseline methods, are taken from their official implementations to maintain fair comparison conditions. Note that we use prior works' original implementations without modification to ensure fair assessment of each method's realistic resource requirements.

**Clarification of SPA.** For SPA, although the paper (Niu et al., 2025) claims that it can generalize to both CNNs and ViTs, the official code provided by the authors includes only the ViT implementation. Consequently, we report comparisons with SPA only on ViT. As shown in Table 1, while SPA achieves competitive performance, it incurs extremely high memory consumption (exceeding 10GB), making it impractical for efficiency-driven applications.

## D    FURTHER DETAILS FOR EVALUATION

### D.1    DETAILED RESULTS ON CIFAR10-C AND CIFAR100-C

While the main paper (Table 2) reports average adaptation performance across the 15 corruption types of CIFAR10-C and CIFAR100-C, we provide the full per-domain performance in this appendix to offer a more granular view of model robustness.

Table 7: Detailed accuracies (%) on CIFAR10-C.

| Model | Methods | gauss. | shot | impul. | defoc. | glass | motion | zoom | snow | frost | fog | brigh. | contr. | elast. | pixel. | jpeg | **Avg.** |
|---|---|---|---|---|---|---|---|---|---|---|---|---|---|---|---|---|---|
| ViT | No Adapt | 43.8 | 49.8 | 69.3 | 82.1 | 71.0 | 82.3 | 84.6 | 91.8 | 89.5 | 87.5 | 95.5 | 83.4 | 83.2 | 51.8 | 81.4 | 76.5 |
| | SAR | 45.4 | 54.7 | 70.0 | 81.9 | 69.8 | 81.7 | 83.9 | 89.0 | 87.5 | 83.6 | 93.3 | 83.2 | 81.7 | 51.5 | 79.5 | 75.8 |
| | Tent | 43.8 | 50.1 | 69.6 | 81.6 | 71.2 | 82.6 | 84.6 | 91.7 | 89.7 | 88.0 | 95.2 | 80.5 | 83.5 | 49.0 | 81.6 | 76.2 |
| | EATA | 43.8 | 49.8 | 69.3 | 82.1 | 71.0 | 82.3 | 84.6 | 91.8 | 89.5 | 87.5 | 95.5 | 83.4 | 83.2 | 51.7 | 81.4 | 76.5 |
| | FOA (F = 27) | 62.7 | 71.6 | 77.5 | 88.2 | 76.4 | 87.7 | 91.1 | 91.6 | 90.9 | 89.1 | 95.2 | 85.3 | 85.2 | 76.5 | 81.3 | 83.3 |
| | FOA (F = 9) | 60.9 | 69.8 | 75.9 | 87.6 | 73.2 | 85.7 | 90.4 | 91.4 | 90.2 | 89.6 | 95.1 | 84.9 | 84.5 | 68.2 | 80.5 | 81.9 |
| | CMF | 66.2 | 71.6 | 71.1 | 87.8 | 75.6 | 87.5 | 90.5 | 89.6 | 90.5 | 88.1 | 94.8 | 86.3 | 83.3 | 83.3 | 83.2 | 83.3 |
| | SPA | 72.3 | 79.2 | 61.8 | 80.3 | 69.8 | 78.9 | 85.7 | 80.4 | 85.2 | 70.1 | 91.5 | 72.7 | 70.1 | 69.9 | 74.9 | 76.2 |
| | PEA | 61.9 | 66.0 | 77.8 | 88.5 | 77.4 | 88.3 | 90.7 | 91.9 | 91.5 | 89.9 | 95.5 | 86.9 | 85.6 | 81.7 | 82.0 | 83.7 |
| | PEA + Aug | 64.5 | 68.3 | 80.2 | 88.6 | 78.8 | 88.4 | 90.4 | 92.8 | 92.6 | 90.6 | 95.8 | 87.1 | 86.5 | 83.1 | 82.6 | **84.7** |
| ResNet | No Adapt | 34.2 | 39.5 | 26.2 | 66.5 | 48.6 | 62.9 | 70.3 | 85.8 | 80.2 | 84.7 | 93.0 | 81.0 | 71.3 | 24.4 | 69.4 | 62.5 |
| | TENT | 63.9 | 68.5 | 65.1 | 89.1 | 67.4 | 87.4 | 90.5 | 87.7 | 86.4 | 90.2 | 93.9 | 91.4 | 80.6 | 83.2 | 72.2 | 81.2 |
| | EATA | 63.9 | 68.4 | 65.1 | 89.3 | 69.3 | 87.4 | 90.5 | 87.4 | 86.8 | 90.4 | 93.8 | 91.4 | 80.8 | 82.9 | 72.0 | 81.3 |
| | MECTA | 64.9 | 69.2 | 64.9 | 89.6 | 69.4 | 88.2 | 92.2 | 88.1 | 87.8 | 90.9 | 94.6 | 92.6 | 81.1 | 83.5 | 73.2 | 82.0 |
| | EcoTTA | 63.9 | 68.0 | 61.8 | 88.7 | 67.4 | 87.3 | 90.6 | 87.3 | 86.7 | 90.8 | 93.9 | 91.5 | 79.5 | 83.1 | 62.9 | 80.2 |
| | L-TTA | 64.4 | 68.9 | 64.0 | 88.7 | 69.3 | 87.1 | 90.3 | 87.4 | 87.0 | 90.0 | 93.3 | 90.5 | 80.6 | 84.0 | 72.9 | 81.2 |
| | CMF | 70.3 | 75.6 | 69.6 | 76.5 | 68.3 | 76.6 | 81.5 | 82.4 | 82.9 | 84.7 | 89.5 | 87.3 | 76.4 | 81.8 | 76.4 | 78.6 |
| | PEA | 65.5 | 69.1 | 73.2 | 88.4 | 69.8 | 86.7 | 91.1 | 89.2 | 88.9 | 91.9 | 94.6 | 93.4 | 80.9 | 78.3 | 73.3 | 82.3 |
| | PEA + Aug | 68.3 | 72.0 | 75.6 | 89.1 | 71.2 | 87.4 | 91.2 | 89.5 | 89.3 | 92.3 | 95.0 | 93.8 | 82.0 | 79.7 | 74.6 | **83.4** |

Table 8: Detailed accuracies (%) on CIFAR100-C.

| Model | Methods | gauss. | shot | impul. | defoc. | glass | motion | zoom | snow | frost | fog | brigh. | contr. | elast. | pixel. | jpeg | **Avg.** |
|---|---|---|---|---|---|---|---|---|---|---|---|---|---|---|---|---|---|
| ViT | No Adapt | 42.0 | 42.4 | 54.1 | 73.1 | 47.5 | 72.8 | 75.0 | 80.5 | 79.7 | 57.1 | 86.0 | 44.0 | 64.5 | 47.7 | 57.5 | 61.6 |
| | SAR | 42.9 | 44.5 | 54.0 | 73.3 | 47.6 | 73.0 | 74.9 | 79.5 | 79.1 | 53.6 | 85.4 | 46.3 | 63.9 | 46.9 | 55.4 | 61.3 |
| | Tent | 42.6 | 43.4 | 54.8 | 73.4 | 47.2 | 73.0 | 75.2 | 80.3 | 79.5 | 54.3 | 86.0 | 39.8 | 64.4 | 46.8 | 57.2 | 61.2 |
| | EATA | 42.5 | 43.8 | 54.5 | 73.2 | 47.9 | 73.0 | 75.0 | 80.1 | 79.7 | 55.6 | 85.7 | 42.0 | 65.1 | 48.9 | 57.2 | 61.6 |
| | FOA (F = 27) | 43.5 | 49.0 | 55.3 | 74.3 | 55.3 | 75.8 | 78.5 | 81.0 | 81.9 | 76.0 | 87.2 | 76.2 | 70.1 | 55.6 | 60.7 | 68.0 |
| | FOA (F = 9) | 44.5 | 48.5 | 52.8 | 73.5 | 50.5 | 74.5 | 78.0 | 80.5 | 80.8 | 75.3 | 87.2 | 75.4 | 70.0 | 55.1 | 61.3 | 67.2 |
| | CMF | 56.5 | 65.3 | 66.0 | 77.0 | 61.1 | 76.4 | 79.5 | 81.0 | 81.7 | 77.8 | 86.1 | 82.9 | 69.6 | 68.7 | 65.7 | 73.0 |
| | SPA | 62.4 | 70.8 | 74.0 | 74.2 | 52.2 | 77.8 | 79.2 | 77.5 | 81.3 | 74.8 | 86.0 | 84.3 | 61.7 | 53.4 | 65.5 | 71.7 |
| | PEA | 62.1 | 64.8 | 71.5 | 81.2 | 65.9 | 80.0 | 82.6 | 82.7 | 83.6 | 82.1 | 87.4 | 86.4 | 70.7 | 70.8 | 64.4 | 75.7 |
| | PEA + Aug | 64.3 | 67.2 | 74.2 | 81.9 | 67.5 | 80.6 | 83.0 | 83.9 | 84.4 | 83.2 | 87.9 | 86.8 | 72.3 | 72.7 | 65.6 | **77.0** |
| ResNet | No Adapt | 12.5 | 14.0 | 8.8 | 34.5 | 16.7 | 36.5 | 42.8 | 53.4 | 46.4 | 52.0 | 68.5 | 39.4 | 35.7 | 9.4 | 32.4 | 33.5 |
| | TENT | 28.9 | 30.7 | 29.7 | 60.4 | 31.9 | 57.1 | 63.2 | 55.6 | 54.1 | 61.2 | 69.5 | 65.4 | 46.4 | 47.1 | 37.4 | 49.2 |
| | EATA | 29.3 | 32.1 | 30.0 | 60.7 | 32.7 | 57.4 | 63.8 | 56.4 | 55.0 | 61.6 | 70.2 | 66.2 | 47.5 | 48.6 | 38.8 | 50.0 |
| | MECTA | 30.2 | 33.9 | 29.3 | 58.4 | 32.3 | 55.7 | 63.1 | 55.7 | 54.9 | 60.8 | 70.0 | 64.4 | 47.3 | 49.2 | 40.5 | 49.7 |
| | EcoTTA | 17.7 | 22.3 | 18.8 | 60.5 | 23.8 | 57.1 | 63.5 | 55.7 | 51.9 | 60.7 | 70.6 | 65.9 | 41.3 | 43.9 | 30.0 | 45.6 |
| | L-TTA | 30.8 | 34.4 | 31.2 | 57.8 | 34.3 | 56.1 | 63.1 | 57.0 | 57.0 | 61.7 | 70.1 | 64.9 | 48.6 | 53.7 | 42.5 | 50.9 |
| | CMF | 38.6 | 44.5 | 37.5 | 45.8 | 37.7 | 46.0 | 53.1 | 52.7 | 53.5 | 53.0 | 63.2 | 58.4 | 46.8 | 54.0 | 46.8 | 48.8 |
| | PEA | 34.3 | 35.7 | 38.2 | 63.4 | 39.1 | 59.8 | 66.8 | 60.1 | 59.8 | 65.7 | 74.4 | 68.4 | 50.0 | 51.6 | 41.1 | 53.9 |
| | PEA + Aug | 35.5 | 36.9 | 39.1 | 64.0 | 39.8 | 60.8 | 67.1 | 60.7 | 60.4 | 66.6 | 74.7 | 68.7 | 51.5 | 52.3 | 41.2 | **54.6** |

Table 7 presents the detailed results on CIFAR10-C. Our method, PEA, consistently outperforms prior baselines across most corruption types for both ViT and ResNet backbones. Notably, PEA + Aug achieves the highest overall accuracy, benefiting from robust alignment and enriched feature diversity. The improvement is especially pronounced under severe corruptions such as impulse noise and pixelate, where domain shifts are more extreme.

Table 8 shows the corresponding breakdown for CIFAR100-C. Similar trends are observed: PEA and its augmented variant deliver consistent gains across nearly all corruption types. On both ViT and ResNet, PEA + Aug achieves the best performance on most corruptions, highlighting the strength of our progressive alignment and augmentation strategy.

These detailed results further demonstrate that our method generalizes well across a wide range of perturbation types, offering both strong average performance and consistent robustness under diverse corruption scenarios.

## D.2 VISUALIZATION OF THE ALIGNED FEATURES

To better understand how our proposed PEA progressively corrects domain shifts across layers, we visualize the intermediate embeddings of three representative classes—pine tree, bicycle, and bee from CIFAR100-C under the Contrast corruption. The upper row of Figure 6 shows the feature distributions of the source (circle markers) and shifted domain samples (triangle markers) at three representative layers (block1, block6, and block11), along with their respective class centroids (stars for source and hexagons for domain). The visualizations clearly reveal that the domain features drift away from the source distributions in all intermediate layers, manifesting as embedding translation, scaling, and rotational shifts.

The lower row presents the domain features after applying our adaptation approach. We observe that the domain clusters become progressively more compact and align closely with the source clusters,

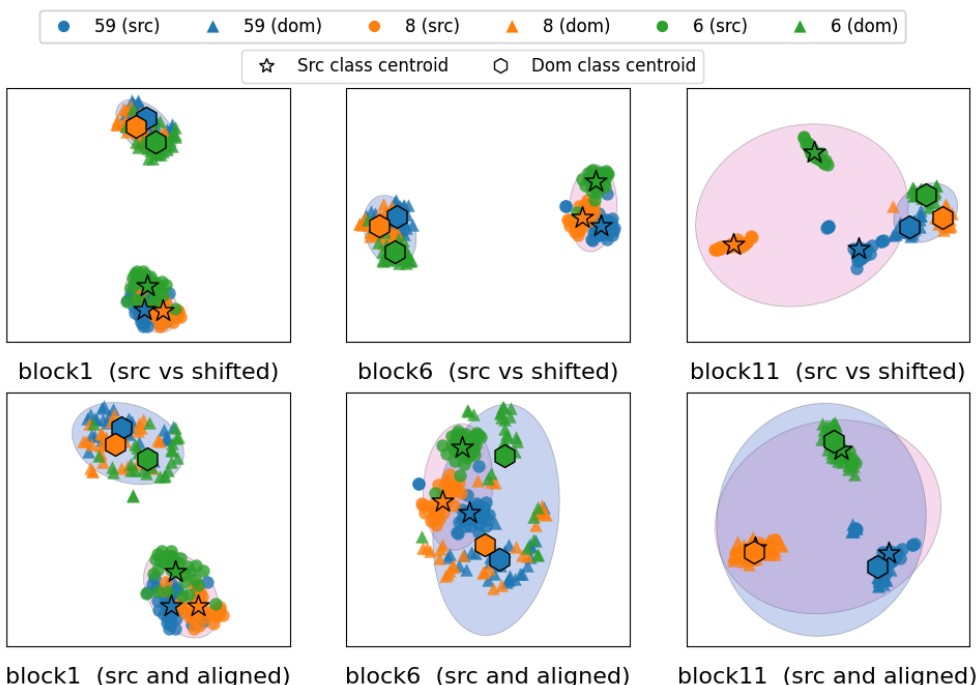

Figure 6: Visualization of intermediate embeddings before and after PEA adaptation on CIFAR100-C. We display the features of the pine tree, bicycle, and bee classes under the CIFAR100-C Contrast corruption.

Table 9: Detailed accuracies (%) under batch size of 4.

| Model | Method | | | | | | | |
|---|---|---|---|---|---|---|---|---|
| ViT-Base | No Adapt | SAR | Tent | EATA | FOA (F = 9) | CMF | SPA | PEA |
| | 55.5 | 63.0 | 62.0 | 62.2 | 62.0 | 62.0 | 62.6 | 63.3 |
| ResNet-50 | No Adapt | TENT | EATA | MECTA | EcoTTA | L-TTA | CMF | PEA |
| | 27.8 | 1.7 | 18.5 | 28.2 | 1.2 | 34.5 | 17.6 | 41.7 |

and the class centroids from the two domains converge. Notably, by the final block (block11), the previously shrinking domain features are largely pulled back to their original source positions. These results demonstrate that PEA can systematically reduce the three types of distortions across and successfully realign the feature spaces to the source distribution.

## D.3 SMALL BATCH SIZE COMPARISON

Table 9 provides a detailed comparison of PEA against existing TTA baselines under a severely constrained setting (batch size = 4). While conventional entropy-minimization methods such as SAR, Tent, and EATA provide consistent improvements around 62%, and FOA (F = 9) reaches similar accuracy, our method further improves performance to 63.3%. For the ResNet-50 backbone, the difference between methods is even more striking. Without adaptation, the baseline model achieves 27.8% accuracy. Most entropy-minimization methods either fail or perform poorly. While MECTA and L-TTA achieve moderate improvements, they still lag behind our method, which attains 41.7%.

The primary reason for the failure on ResNet is that most existing TTA baselines rely on updating BatchNorm (BN) statistics. This process requires sufficiently large batch sizes to estimate stable mean and variance values; otherwise, the updates become noisy and cause severe performance degradation. Under small-batch regimes, such as those common on edge devices, these baselines therefore collapse in accuracy. In contrast, PEA avoids this limitation by not depending on BN updates or backpropagation. Instead, it realigns embeddings using pre-computed source statistics and lightweight covariance alignment, which remain stable even with very small batches. This design

Table 10: Results on BS=1 and BS=2 on CIFAR100-C and ImageNet-C with ViT-Base.("F" indicates that the method fails under BS=1.)

| Method | ImageNet-C | | CIFAR100-C | |
|---|---|---|---|---|
| | Batch Size = 1 | Batch Size = 2 | Batch Size = 1 | Batch Size = 2 |
| No Adapt | 55.5 | 55.5 | 61.6 | 61.6 |
| SAR | 58.6 | 62.0 | 62.0 | 64.2 |
| CMF | F | 61.5 | F | 61.5 |
| FOA (K=9) | F | 64.4 | F | 66.9 |
| SPA | 59.5 | 60.8 | 46.5 | 70.6 |
| **PEA** | **61.6** | **62.5** | **69.5** | **69.9** |

Table 11: Effect of momentum $m$ and entropy threshold $\theta_{\text{ent}}$ on average accuracy (%) on CIFAR100-C. Each cell reports accuracy with the average number of entropy spikes in ($\cdot$).

| $m$ | Accuracy (%) (Spikes) vs. $\theta_{\text{ent}}$ | | | |
|---|---|---|---|---|
| | 0.5 | 0.8 | 1.0 | 1.5 |
| 0.01 | 75.3 (16) | 75.4 (9) | 75.0 (7) | 74.6 (2) |
| 0.02 | 75.7 (17) | 75.8 (9) | 75.5 (5) | 75.5 (2) |
| 0.05 | 75.7 (17) | 75.7 (6) | 75.7 (3) | 75.7 (2) |
| 0.10 | 75.4 (17) | 75.3 (8) | 75.4 (2) | 75.3 (1) |

makes PEA inherently more robust under constrained batch sizes, ensuring consistent performance across both CNN and ViT backbones.

## D.4 STREAMING EVALUATION WITH EXTREMELY SMALL BATCHES (BS=1/2)

To reflect a realistic streaming deployment where test inputs arrive one (or a few) at a time, we evaluate all methods under extremely small batch sizes (BS=1 and BS=2). This regime is particularly challenging for test-time adaptation because reliable batch statistics are difficult to estimate and some methods become numerically unstable or memory-inefficient when the batch degenerates to a single sample. Table 10 summarizes the results on ImageNet-C and CIFAR100-C with ViT-Base.

PEA remains stable and consistently improves over the non-adapted source model in both datasets, achieving strong performance even at BS=1 (e.g., 61.6 on ImageNet-C and 69.5 on CIFAR100-C). In contrast, several baselines fail at BS=1 (e.g., CMF and FOA), highlighting their reliance on larger batches or iterative test-time optimization. These results validate PEA's suitability for streaming test-time inference: by performing lightweight, statistics-driven progressive alignment without backpropagation, PEA can operate reliably when only minimal batch information is available.

## D.5 MORE RESULTS OF ABLATION STUDY

### D.5.1 HYPERPARAMETERS

We further analyze the sensitivity of our method to the two hyperparameters: the EMA momentum $m$, which controls the update rate of domain statistics, and the entropy threshold $\theta_{\text{ent}}$, which is used for detecting distribution shifts. To this end, we sweep a range of values for both parameters and report the average accuracy across the 15 CIFAR100-C domains, together with the number of entropy spikes triggered during evaluation, as shown in Table 11.

The results show that very small momenta (e.g., $m = 0.01$) underperform with accuracies around 75.0–75.4%, while moderate values of $m = 0.02$–$0.05$ consistently achieve the best performance (75.5–75.8%). Larger momentum ($m = 0.10$) again reduces accuracy to about 75.3–75.4%. For the entropy threshold, low values such as $\theta_{\text{ent}} = 0.5$ lead to frequent resets (16–17 spikes), whereas high values like $\theta_{\text{ent}} = 1.5$ almost disable resets (1–2 spikes). Accuracy remains stable across thresholds with differences within 1%, but excessively low thresholds slightly degrade performance due to too many resets, while overly high thresholds risk ignoring meaningful shifts. The balanced setting of $\theta_{\text{ent}} = 0.8$–$1.0$ achieves both high accuracy (75.5%–75.8%) and moderate spike counts.

Table 12: Impact of the fraction of source data used to adaptation accuracy(%).

| Source Data Used | CIFAR10-C ViT | CIFAR100-C ViT |
|---|---|---|
| 5% | 83.2±0.1 | 76.4±0.2 |
| 10% | 83.2±0.1 | 76.8±0.2 |
| 20% | 83.4±0.1 | 77.0±0.1 |
| 50% | 83.4±0.1 | 77.0±0.1 |
| 100% | 83.4±0.1 | 77.0±0.1 |

Table 13: Impact of the number of test-time augmentation views K on top-1 accuracy (%).

| Dataset | K = 0 | K = 1 | K = 2 | K = 3 | K = 4 |
|---|---|---|---|---|---|
| ImageNet-C | 64.5 | 66.2 | 66.5 | 66.6 | 66.6 |
| CIFAR100-C | 75.7 | 76.6 | 77.0 | 77.0 | 77.1 |

In conclusion, our method is not highly sensitive to these hyperparameters, with accuracy variation contained within about 1%. For all main experiments, we adopt $m = 0.02$ and $\theta_{\text{ent}} = 1.0$ as they provide the best trade-off between responsiveness and stability.

### D.5.2  SENSITIVITY TO SOURCE DATA SIZE FOR OFFLINE STATISTICS

PEA requires access to a small amount of source data **before deployment** to estimate and store per-block source statistics. To understand how much source data is needed in practice, we vary the fraction of source samples used in this offline stage and evaluate the resulting adaptation accuracy. As shown in Table 12, PEA is largely insensitive to the amount of source data used to compute the offline statistics. On CIFAR10-C, performance saturates quickly even with 5% of source data. On CIFAR100-C, using only 10% of source samples already achieves near-saturated performance (76.8% vs. 77.0% with 100%), and 20% matches the full-data result. Importantly, this preprocessing is performed once and entirely offline; at test time, PEA only relies on the stored statistics and never accesses source data again.

### D.5.3  NUMBER OF VIEWS

In our PEA, the data augmentation will add more views ($K$) of the data to capture more accurate statistics. Our default setting in the paper is $K = 2$. We add more ablation on the number of augmentation views from $K = 0$ to $4$ in the following Table 13. We observe that multi-view augmentation consistently improves performance over $K = 0$ (no augmentation), with the largest gain from $K = 0$ to $K = 1$, and a smaller but still clear gain from $K = 1$ to $K = 2$. Beyond $K = 2$, the improvements saturate and become marginal (¡0.1%). Since increasing K also increases compute and memory, we adopt $K = 2$ as the default to capture most of the benefit of augmentation while keeping the overhead modest.

## E  DOMAIN SHIFT IN RESNET EMBEDDING SPACE

In this section, we illustrate that the embedding-space shift for ResNet is consistent with that observed in ViT. We visualize the embedding space of ResNet-50 on CIFAR10-C and observe the same systematic shifts in mean, variance, and channel-wise covariance across layers. Specifically, Figure 7 and Figure 8 show the Zoom Blur and Frost domains, respectively, where the embedding shifts closely match those seen in ViT, indicating that the effect is driven by domain shift rather

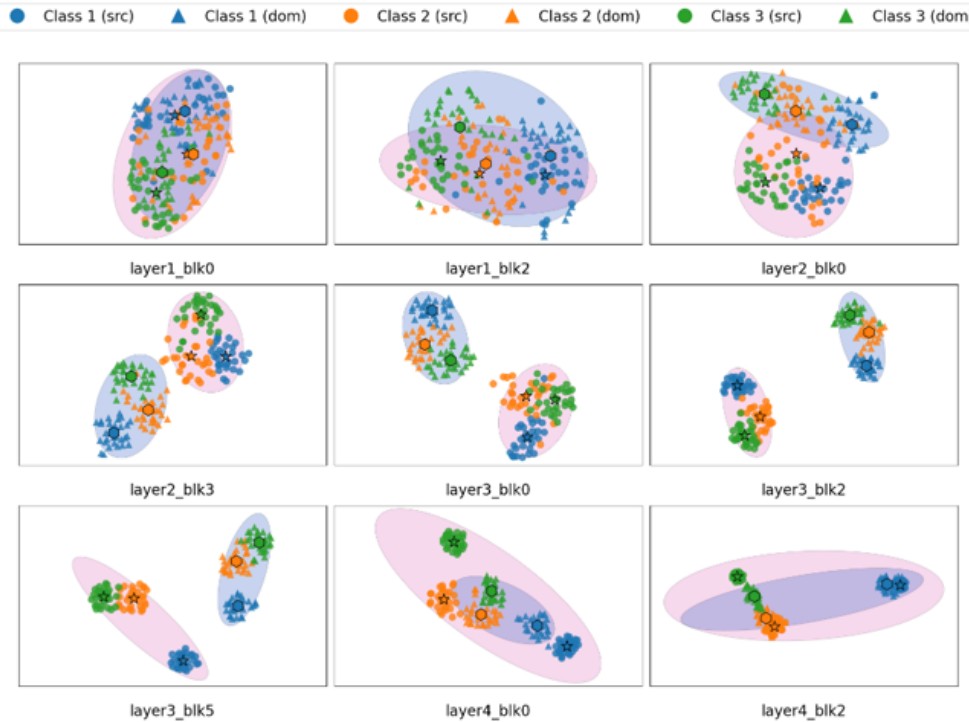

Figure 7: Impact of domain shift on intermediate layer embeddings using **CIFAR100-C**. We visualize the features in Domain **Zoom Blur** for **ResNet-50**.

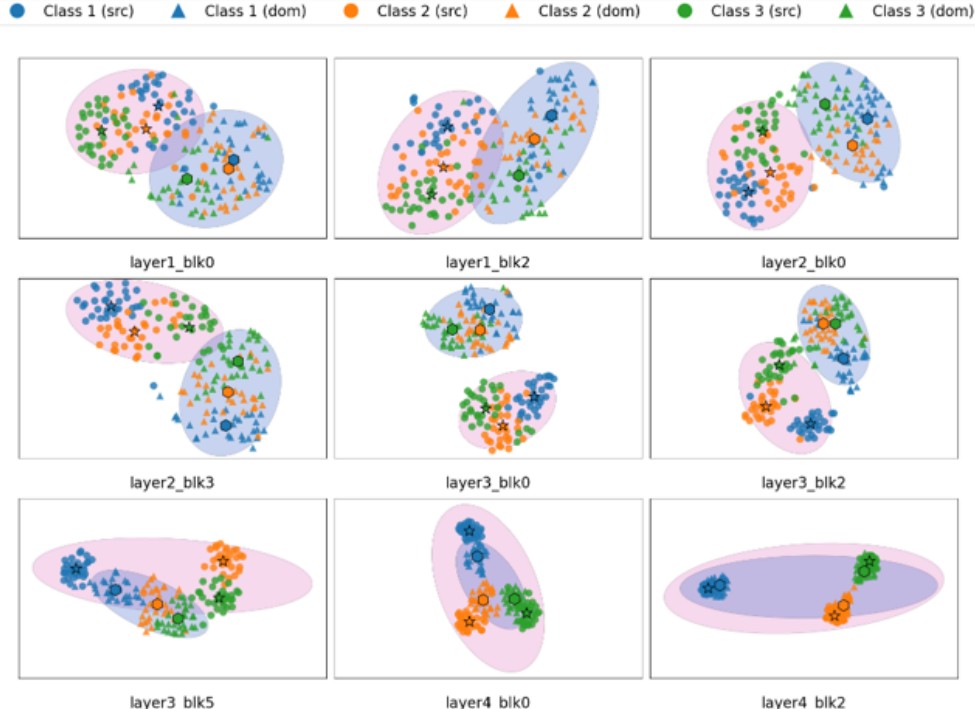

Figure 8: Impact of domain shift on intermediate layer embeddings using **CIFAR100-C**. We visualize the features in Domain **Frost** for **ResNet-50**.

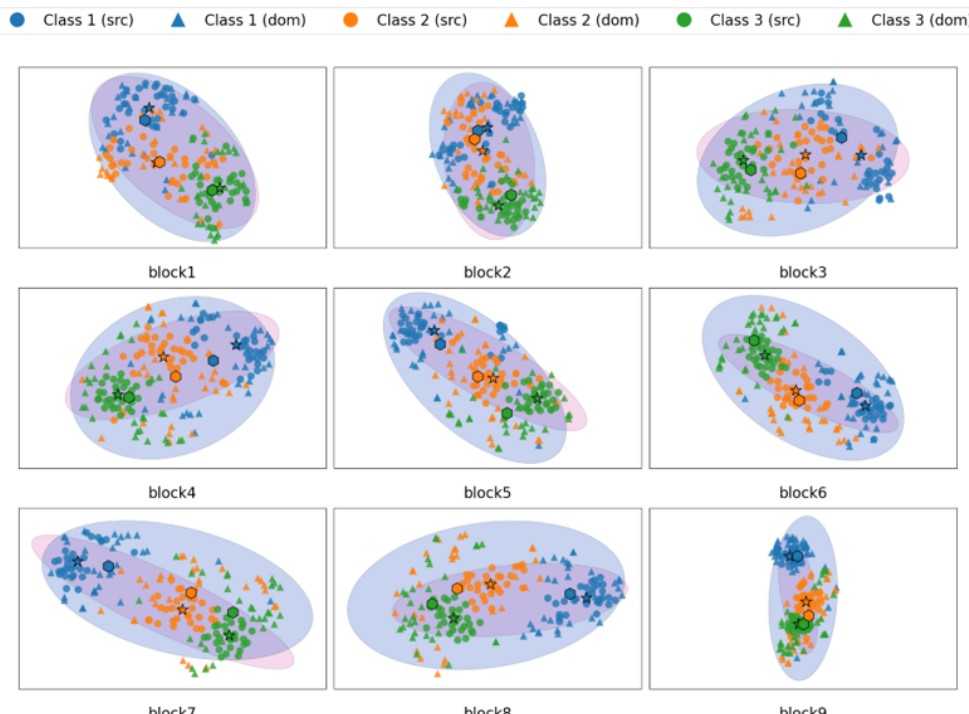

Figure 9: Impact of mixed domains on intermediate-layer embeddings on **CIFAR100-C**. We visualize ViT-Base features for 3 classes.

than architectural design. This aligns with our experimental results, where PEA also improves CNN backbones (ResNet-50) on CIFAR10-C/100-C and ImageNet-C.

## F    MIXED-DOMAIN EMBEDDING SHIFT VISUALIZATION

To better understand why PEA remains effective under the mixed-domain scenario, we visualize the intermediate embeddings of ViT-Base on CIFAR100-C in a mixed-domain setting. Specifically, we construct a mixed CIFAR100-C stream where all 15 corruptions (severity 5) are combined and shuffled, so that each batch contains samples from multiple domains. For a subset of 3 classes, we extract features from 9 ViT blocks (from shallow to deep) and jointly project the clean and mixed-domain embeddings. Figure 9 shows the resulting embeddings for all 9 blocks.

As expected, compared to the clean case, mixed-domain embeddings exhibit lower intra-class compactness and increased spread, since each class now aggregates samples from heterogeneous corruptions. However, the key observation is that the distortion is still highly systematic across layers. This matches our hypothesis that domain shift, even when composed of multiple human-defined "domains" (fog, snow, blur, etc.), induces a geometric shift in the embedding space.

## G    USE OF LLM IN THIS PAPER

We emphasize that large language models (LLMs) were used solely for polishing the writing and improving readability. No part of the technical content, experimental design, analysis, or results relied on LLM-generated material. All research ideas, implementations, and evaluations are original to the authors.

