# OpenReview forum: "Architecture-Agnostic Test-Time Adaptation via Backprop-Free Embedding Alignment"
_ICLR.cc/2026/Conference — ICLR 2026 Poster_

### Official Review · Reviewer_Lsas · 2025-10-29

**Soundness:** 3
**Presentation:** 2
**Contribution:** 2
**Rating:** 4
**Confidence:** 4

**Summary:**

The authors propose a new backpropagation-free test-time adaptation method to calibrate the pre-trained model to the target domains using two forward passes per sample. The authors observe that the distribution shift can cause three distinct structural changes in the features of intermediate layers. Based on this observation, they design a Progressive Embedding Alignment (PEA) to correct the feature distortions. One forward is used to estimate the level of distribution shift, and the other is used to recompute the intermediate features to tackle the distribution shift. Moreover, they also use the exponential moving average and data argumentation to enable stable and robust adaptation. Experimental results demonstrate the effectiveness of the proposed method.

**Strengths:**

1. The authors empirically reveal that the distribution shift can cause the three distinct structural changes in the feature space, including translation (mean shift), scaling (variance shift), and rotation (covariance shift).
2. The authors propose a Progressive Embedding Alignment (PEA) to calibrate the feature without updating the parameters of the model, which can originally mitigate catastrophic forgetting.
3. The results of the ablation study demonstrate the necessity of each component in the proposed method.

**Weaknesses:**

1. The authors propose a feature re-computation method to calibrate the pre-trained model. However, a similar idea that updates the intermediate features has been proposed by the previous method[A]. What is the essential difference and advantage of the proposed method?
2. The authors show the distinct structural changes of the ViT model in Figure 1. However, it is unclear whether this observation is related to this specific architecture. More discussion or experimental results on CNN models or models with different normalization layers should be provided.
3. The proposed method requires computing the statistics of a batch of test samples, so the distribution of different domains in the same batch will greatly affect the accuracy of the statistical distribution. When the test samples within the same batch come from different domains (mixed data domains), the proposed method may suffer from unreliable distribution estimation and cannot guarantee a stable performance.
4. The Spike detection is proposed to detect the domain changes and then reset the estimated statistical information for the new domain. However, it is unclear how to decide the threshold \theta_{ent} for different domains, especially under challenging test scenarios such as imbalanced label shifts per SAR.
5. The data enrichment is designed to enhance the estimation of the target batch distribution by generating K augmented views for each test sample. However, it is unclear how the hyperparameter K affects the adaptation performance and the computational cost.
6. More BP-free test-time adaptation methods should be discussed and compared in the experiment, including but not limited to [B-D].

**Questions:**

See Weaknesses.

---

> ### Author Response · Authors · 2025-11-20
> **Response to Reviewer Lsas (Part 1)**
>
> > **Comment 1:** The authors propose a feature re-computation method to calibrate the pre-trained model. However, a similar idea that updates the intermediate features has been proposed by the previous method\[A\]. What is the essential difference and advantage of the proposed method?
>
> We thank the reviewer for pointing out LeanTTA \[A\]. Conceptually, both LeanTTA and PEA are backpropagation-free TTA methods, but they do so in **very different ways**. LeanTTA adapts by **updating normalization statistics** in the BatchNorm layers through a weighted combination of source and test statistics. This implicitly changes intermediate features but is **tightly coupled to the presence of BN layers**, and is therefore only applicable to CNN-style or MobileViT architectures with BN layers. In contrast, PEA does not rely on any specific layer type. It performs explicit feature alignment across layers at the embedding level. This makes PEA **architecture-agnostic** and directly applicable to both **ResNet** and **ViT** backbones. In other words, LeanTTA still modifies the pre-trained model weights to fit the new domain, while PEA aligns the distorted embeddings to fit the pre-trained model.
>
> > **Comment 2:** The authors show the distinct structural changes of the ViT model in Figure 1\. However, it is unclear whether this observation is related to this specific architecture. More discussion or experimental results on CNN models or models with different normalization layers should be provided.
>
> Thank you for the suggestion. The phenomenon is **not** ViT-specific. We replicated the shift in Fig. 1 for ResNet-50 on CIFAR10-C and observed the same systematic shifts in mean, variance, and channel-covariance across layers. We visualize two domains on ResNet-50 in Fig. 7 and Fig. 8 in the revised paper (please see Appendix F, pages 19 and 20), and the embedding shifts closely match those seen in ViT, indicating that the effect is caused by domain shift rather than architectural design. This is consistent with our experiment results in the paper, where PEA also improves CNN backbones (ResNet-50) on CIFAR10-C/100-C and ImageNet-C.
>
> Note that PEA aligns intermediate embeddings across layers without updating any parameters, and is therefore agnostic to the underlying normalization layer (ViT uses LayerNorm; ResNet uses BatchNorm). This contrasts with many mainstream TTA methods (e.g., Tent, EATA, SAR), whose reliance on normalization-layer updates limits their architectural generality. We have added these ResNet visualizations and a brief discussion of this point to Appendix F in the revised paper.
>
> (Due to space limitations, we present the response to Comment 4 here; please refer to Part 2 for our response to Comment 3.)
> > **Comment 4:** The Spike detection is proposed to detect the domain changes and then reset the estimated statistical information for the new domain. However, it is unclear how to decide the threshold \\theta\_{ent} for different domains, especially under challenging test scenarios such as imbalanced label shifts per SAR.
>
> We use the prediction entropy for spike detection because it is a simple and direct indicator of prediction uncertainty. When a domain shift occurs, this uncertainty typically **spikes** due to the mismatch between source-trained decision boundaries and shifted inputs. The threshold is set to 1 in the paper based on our empirical analysis. Specifically, we analyzed the prediction entropy across batches and found that the entropy usually ranges from 0 to 0.8 in the training set. Therefore, we empirically set the threshold as 1\. To evaluate the impact of the threshold selection, we added additional experiments with different threshold values during the rebuttal. As shown in Table R4-2, the impact on accuracy is marginal. This is because (1) shift detection is mainly valid at the boundary of the domain transitions and (2) as long as the threshold can differentiate two domains, the performance will not be affected.
>
> Table R4-2. Impact of the shift detection threshold on accuracy (%).
>
> | Dataset | 0.5 | 1.0 | 1.5 | 2.0 |
> | :---: | :---: | :---: | :---: | :---: |
> | ImageNet-C | 66.4 | 66.5 | 66.4 | 66.4 |
> | CIFAR100-C | 77.0 | 77.0 | 77.1 | 76.8 |

---

> ### Author Response · Authors · 2025-11-20
> **Response to Reviewer Lsas (Part 2)**
>
> > **Comment 3:** The proposed method requires computing the statistics of a batch of test samples, so the distribution of different domains in the same batch will greatly affect the accuracy of the statistical distribution. When the test samples within the same batch come from different domains (mixed data domains), the proposed method may suffer from unreliable distribution estimation and cannot guarantee a stable performance.
>
> Thanks for this insightful comment\! Our main experiments follow the standard continual single-domain protocol used in prior CTTA works, but we do agree that the mixed-domain batch is a more challenging case and worth exploring.
>
> First, we constructed a mixed CIFAR100-C dataset where all 15 corruptions (severity 5\) are combined into a single pool and randomly shuffled, i.e., each batch contains samples from multiple domains. This ensures that each batch of 64 samples contains 4.3 samples on average per corruption type. This represents a realistic scenario where the model encounters rapid, unpredictable domain shifts within individual batches, which is more challenging than the single-domain setting where each batch consists of samples from a single corruption type.
>
> As shown in Table R4-1, PEA with ViT achieves **72.0%**, a substantial improvement over the **61.6%** source model, and outperforms all the baselines.
>
> Table R4-1. Results on mixed-domain data on CIFAR100-C with ViT-Base.
>
> | Dataset | Source Model | Tent | EATA | SAR | FOA (K=9) | CMF | SPA | PEA |
> | :---: | :---: | :---: | :---: | :---: | :---: | :---: | :---: | :---: |
> | CIFAR100-C | 61.6 | 61.2 | 61.2 | 62.0 | 62.6 | 69.0 | 71.4 | 72.0 |
>
> To understand why PEA remains effective, we visualize the intermediate embeddings for 9 ViT blocks using t-SNE, comparing **clean data vs. mixed-domain data** (please see Appendix G, Fig. 9 in the revised paper). As expected, mixed-domain batches exhibit **lower intra-class compactness** because samples originate from heterogeneous domains. However, the key observation is that the **structural distortion is still systematic** across blocks: class clusters undergo consistent **mean shift**, **variance shift**, and **channel-wise covariance shift**.
>
> Since PEA aligns features via a WCT and applies **layer-wise distance-aware weighting** together with **EMA smoothing**, it corrects these dominant geometric shifts even when the batch distribution is a mixture domain. This explains why PEA continues to improve accuracy significantly under mixed-domain TTA.
>
> Additionally, we want to clarify that ‘mixed-domain’ actually refers to the semantic characteristics defined by humans (e.g., fog, snow). However, at the embedding space, there might be another high-dimensional space where these mixed domains exhibit similar features. As a result, TTA approaches are still able to find a proper perspective for adaptation or alignment.
>
> > **Comment 5:** The data enrichment is designed to enhance the estimation of the target batch distribution by generating K augmented views for each test sample. However, it is unclear how the hyperparameter K affects the adaptation performance and the computational cost.
>
> We thank the reviewer for the suggestion. Our default setting in the paper is K=2 and during rebuttal, we added another ablation on the number of augmentation views from K=0 to 4 in the following table R4-3. We observe that multi-view augmentation consistently improves performance over K=0 (no augmentation), with the largest gain from K=0 to K=1, and a smaller but still clear gain from K=1 to  K=2. Beyond K=2, the improvements saturate and become marginal (\<0.1%). Since increasing K also increases compute and memory, we adopt **K=2** as the default to capture most of the benefit of augmentation while keeping the overhead modest.
>
> Table R4-3. Impact of the number of test-time augmentation views K on accuracy (%).
>
> | Dataset | K \= 0 | K \= 1 | K \= 2 | K \= 3 | K \= 4 |
> | :---: | :---: | :---: | :---: | :---: | :---: |
> | ImageNet-C | 64.5 | 66.2 | 66.5 | 66.6 | 66.6 |
> | CIFAR100-C | 75.7 | 76.6 | 77.0 | 77.0 | 77.1 |

---

> ### Author Response · Authors · 2025-11-20
> **Response to Reviewer Lsas (Part 3)**
>
> > **Comment 6:** More BP-free test-time adaptation methods should be discussed and compared in the experiment, including but not limited to \[B-D\].
>
> Thanks for the references. Since the methods in C and D are not publicly available, we only report comparisons against T3A \[B\] in Table R4-4 below. Because C and D are unreleased during the submission, we did not include them in our related work, and we initially omitted T3A as a baseline since we had already selected FOA as the state-of-the-art backprop-free baseline. With the newly added results, we can observe that PEA outperforms T3A by **9.6%** on accuracy. We will include T3A as an additional baseline and report these results in the final version.
>
> Table R4-4. Average Accuracy on ImageNet-C compared with FOA and T3A.
>
> | Dataset | No Adapt | FOA (F \= 9\) | T3A | PEA |
> | :---: | :---: | :---: | :---: | :---: |
> | ImageNet-C | 55.5 | 64.3 | 56.9 | 66.5 |
>
> For the newly proposed methods ZOA \[C\] and ADAPT \[D\], we briefly discuss their relation to PEA. ZOA uses a perturbation-based zeroth-order optimizer to *update* a subset of model parameters at test time. In contrast, PEA **never updates** model parameters: we deliberately keep the pretrained backbone fixed, under the assumption that it already encodes strong task knowledge, and instead adapt by transforming the **features** via closed-form statistics-based alignment. This design helps PEA remain stable in challenging settings such as **mixed-domain streams** and **very small batch sizes**, where repeated parameter updates can easily become unstable.
>
> ADAPT, on the other hand, models **class-conditional Gaussians in CLIP feature space** using a vision–language model, and therefore **requires a text encoder and class text embeddings** as an essential component of the method. It is tailored to CLIP-style VLMs. By contrast, PEA targets the more standard setting with **general CNN/ViT classifiers** (no text encoder) and performs **backprop-free** embedding alignment under the classic TTA setup. We will clarify these complementary scopes and methodological differences in the revised version.
>
>
> **Summary:​**
>
> We have thoroughly addressed your concerns by: (1) clarifying the essential differences from LeanTTA (BN-specific vs. architecture-agnostic embedding alignment); (2) adding ResNet-50 visualizations demonstrating the same ​s​hifts occur in CNNs (Figures 7 and 8 in Appendix F in revised paper); (3) conducting mixed-domain experiments showing 72.0% accuracy vs. 61.6% baseline (Table R4-1); (4) providing threshold sensitivity analysis showing robustness (Table R4-2); (5) adding augmentation ablation demonstrating K=2 is optimal (Table R4-3); and (6) comparing with T3A and discussing ZOA/ADAPT, with PEA outperforming T3A by 9.6% (Table R4-4). We believe these new results and clarifications substantially strengthen the contribution and generality of our work.

---

### Official Review · Reviewer_9W85 · 2025-10-30

**Soundness:** 2
**Presentation:** 3
**Contribution:** 2
**Rating:** 2
**Confidence:** 3

**Summary:**

This work proposes PEA, a backpropagation-free and architecture-agnostic test-time adaptation (TTA) approach that aims to enhance adaptation while reducing computational bottlenecks. Unlike most TTA methods that rely on backpropagation—which is computationally expensive at test time—or are architecture-specific, PEA provides a general alternative. The authors observe that the representations of shifted domain samples across layers are geometric transformations of the source representations, primarily translation, scaling, and rotation. Building on this insight, PEA performs two forward passes: the first collects batch statistics per layer, and the second applies a correction based on source statistics. Experimental evaluations on ResNet-50 and ViT-B across corrupted datasets (CIFAR-10-C, CIFAR-100-C, and ImageNet-C) demonstrate accuracy gains when combined with test-time augmentation. Additional experiments on low-resource devices further highlight the method’s memory efficiency.

**Strengths:**

* The paper is well written and easy to follow.

* The paper proposes a method with two interesting properties: backpropagation-free and architecture-agnostic, which increases the applicability of TTA across different models and devices.

* The evaluation on low-resource devices is a valuable contribution, highlighting the method’s efficiency in practical settings.

**Weaknesses:**

* The paper emphasizes the existence of discrepancies in representations between source and target distributions and their variation across layers. However, this adds limited new understanding. By definition, distribution shifts alter centroids and other statistical descriptors, and much of the domain adaptation literature already focuses on aligning source and target distributions.

* The hypothesis that shifts reduce to simple geometric transformations in embedding space is overly strong, and the visualizations in Figure 1 and Appendix A do not provide convincing evidence. For example, as long as representations differ, a translation between centroids is always possible and therefore not surprising. Moreover, these are linear operations, and evidence from only two transformations is insufficient to claim generality. Non-linear or more complex transformations may well occur in intermediate representations, and the current evidence does not rule this out.

* A critical assumption is that source statistics for all layers are accessible at test time. This may not hold in many realistic scenarios and sets the method apart from approaches that do not make such an assumption. This should be acknowledged explicitly as a limitation.

* The conclusions in lines 160–163 and 139-141 overlook prior fine-grained analyses, such as Sahoo et al. and Maharana et al., which emphasize that not all layers are equally affected and advocate for layer-wise adaptation.


* The detection mechanism based on the entropy of predictions is limited. Entropy is a weak OOD detection score compared to alternatives available in the literature (e.g., OpenOOD benchmark).

References

Sahoo, Sabyasachi, et al. "A layer selection approach to test time adaptation." Proceedings of the AAAI Conference on Artificial Intelligence. Vol. 39. No. 19. 2025.

Maharana, Sarthak Kumar, Baoming Zhang, and Yunhui Guo. "Palm: Pushing adaptive learning rate mechanisms for continual test-time adaptation." Proceedings of the AAAI Conference on Artificial Intelligence. Vol. 39. No. 18. 2025.

OpenOOD benchmark: https://github.com/Jingkang50/OpenOOD

**Questions:**

* Class representations appear already well separable after only 3 blocks (out of 9). How do you explain this observation?

* Line 167: “This systematic shift does not lead to a random feature distortion; rather, it manifests as a geometric rotation and shearing of the entire feature cloud.” Why would we expect a random feature distortion in the first place?

* In pass1 paragraph of line 290: How are K views of multiple augmentations obtained within a single forward pass? Please clarify this point.

* For clarity, in the main text, please specify what a “block” corresponds to in the ViT architecture, and explicitly indicate that the referenced block 3 is the third out of nine blocks.

* Please cite prior work using test-time augmentation (e.g., Simonyan and Zisserman, 2014).

Reference

Simonyan, Karen, and Andrew Zisserman. Very deep convolutional networks for large-scale image recognition. arXiv preprint arXiv:1409.1556 (2014).

---

> ### Author Response · Authors · 2025-11-20
> **Response to Reviewer 9W85 (Part 1)**
>
> > **Comment 1:** The paper emphasizes the existence of discrepancies in representations between source and target distributions and their variation across layers. However, this adds limited new understanding. By definition, distribution shifts alter centroids and other statistical descriptors, and much of the domain adaptation literature already focuses on aligning source and target distributions.
>
> We agree that our work is related to distribution alignment methods such as Deep CORAL \[1\] in the traditional domain adaptation literature, but methodologically, it is quite different. Prior alignment-based approaches treat covariance alignment as a **loss term** and rely on **updating model parameters (or an explicit transformation module) via backpropagation**, so that the network *learns* a mapping that reduces the CORAL loss between source and target features during training.
> In contrast, PEA **never updates the model**. The backbone remains completely frozen, and we apply **explicit** embedding alignment **at test time**. Moreover, PEA operates under the **TTA setting**, where only a streaming, unlabeled target batch is available, without access to the full target domain as in standard UDA. This is a more restrictive and practical scenario. In other words, CORAL-style methods **optimize the model to fit the features**, whereas PEA **optimizes the features to fit a fixed model** in a fully backprop-free, test-time setting. We will clarify this conceptual and methodological difference in the final paper.
>
> \[1\] Sun B, Saenko K. Deep coral: Correlation alignment for deep domain adaptation\[C\]//European conference on computer vision. Cham: Springer International Publishing, 2016: 443-450.
>
> > **Comment 2:** The hypothesis that shifts reduce to simple geometric transformations in embedding space is overly strong, and the visualizations in Figure 1 and Appendix A do not provide convincing evidence. For example, as long as representations differ, a translation between centroids is always possible and therefore not surprising. Moreover, these are linear operations, and evidence from only two transformations is insufficient to claim generality. Non-linear or more complex transformations may well occur in intermediate representations, and the current evidence does not rule this out.
>
> We appreciate this comment and agree that our original hypothesis was too strong. We do not claim that domain shift *fully reduces* to simple linear transformations or that more complex, non-linear effects are absent. Our empirical finding is that, across many layers on both CNN and ViT backbones, the **dominant changes in the embeddings are captured by first- and second-order statistics** (mean, variance, and channel-wise covariance), which can be systematically corrected by our efficient alignment procedure.
>
> At the same time, we acknowledge that there are **local and more complex distortions,** including inter-class overlap and intra-class dispersion that are not well described by simple linear transformations. Our claim is therefore more modest: despite these local effects, we consistently observe a **systematic global shift** in mean/variance/covariance across layers, and correcting this dominant component already yields substantial gains in practice. We will soften the phrasing in the revised paper rather than suggesting that all domain shifts are strictly linear.
>
> > **Comment 3:** A critical assumption is that source statistics for all layers are accessible at test time. This may not hold in many realistic scenarios and sets the method apart from approaches that do not make such an assumption. This should be acknowledged explicitly as a limitation.
>
> We agree with the reviewer and explicitly acknowledged this limitation in the original submission (Line 484-485). There are indeed some methods (like Tent, SAR) that are completely source-free. However, many recent TTA works (e.g.,  EcoTTA, EATA, FOA, L-TTA) that yield better performance and efficiency also require source data for one-time offline pre-computing, similar to our work, PEA.
>
> To alleviate this issue, we conducted an additional experiment to evaluate the impact of source data size. The result indicates that as few as 10% of source samples are sufficient to obtain accurate statistics for CIFAR100 as Table R3-1 shows. Moreover, this preprocessing is performed **once and entirely offline** before deployment; at test time, PEA only uses the stored statistics and never accesses source data again. We will add the discussion of the impact of source data size in the camera-ready manuscript.
>
> Table R3-1. Impact of the fraction of source data used to estimate statistics on CIFAR100-C performance with PEA.
>
> | Source Data Used | Accuracy (%) |
> | :---: | :---: |
> | 10% | 76.8 |
> | 20% | 77.0 |
> | 100% | 77.0 |

---

> ### Author Response · Authors · 2025-11-20
> **Response to Reviewer 9W85 (Part 2)**
>
> > **Comment 4:** The conclusions in lines 160–163 and 139-141 overlook prior fine-grained analyses, such as Sahoo et al. and Maharana et al..
>
> We thank the reviewer for pointing this out. We have carefully read GALA and PALM and agree with their observation that **not all layers are equally affected by adaptation**, and that one can use **gradient-based statistics** to perform layer selection to decide *which* layers to update and *how strongly* during backprop-based TTA.
>
> In fact, PEA also adopts a similar strategy for layer-wise alignment. Specifically, the first pass in PEA analyzes the magnitude of embedding shift across different layers and assigns different weights for alignment in the second pass. The layers that are less affected by domain shift are therefore assigned with smaller weights, similar to selectively layer updates in GALA. We have cited these works in the revised version.
>
> > **Comment 5:** The detection mechanism based on the entropy of predictions is limited. Entropy is a weak OOD detection score compared to alternatives available in the literature.
>
> We agree that entropy is not the strongest OOD score in the dedicated OOD detection literature (e.g., OpenOOD), where more sophisticated scores such as ODIN often perform better. However, our setting is different: we do not aim to solve a full open-set OOD detection problem, but only to **detect significant domain shift in a continual online setting** in order to reset the EMA statistics. For this purpose, we deliberately choose a **simple, cheap, and model-agnostic** metric. Entropy is already widely used in TTA as a proxy for uncertainty and shift (e.g., Tent, EATA, SAR), and it can be computed from the logits with **negligible additional cost**.
>
> > **Comment 6:** Class representations appear already well separable after only 3 blocks (out of 9). How do you explain this observation?
>
> We agree that in Fig. 1 the visualized classes already look fairly separable after 3 blocks. This is due to how we construct the figure: to make the t-SNE plots readable and highlight the *structural effect* of domain shift, we visualize only a small subset of 3 classes (out of 10), which are relatively easy to separate. However, this does not mean that early layers are sufficient for the full problem. When we consider **all classes**, especially more confusing ones, deeper blocks are still crucial to refine decision boundaries, increase margins, and maintain robustness under stronger corruptions. We will clarify in the text that the visualization is based on a small, illustrative class subset chosen for clarity.
>
> > **Comment 7:** Line 167: “This systematic shift does not lead to a random feature distortion; rather, it manifests as a geometric rotation and shearing of the entire feature cloud.” Why would we expect a random feature distortion in the first place?
>
> We thank the reviewer for the clarification. Our intention was not to suggest that random distortion is expected, but to emphasize that the observed shift is **highly systematic**. We have revised the sentence in the revised version to focus on this notion of systematic geometric shift and avoid implying that randomness is the baseline expectation.
>
> > **Comment 8:** In pass1 paragraph of line 290: How are K views of multiple augmentations obtained within a single forward pass? Please clarify this point.
>
> In Pass 1, the K augmented views are produced before the forward pass and concatenated with the original batch, giving an effective input batch size of (K+1) that is processed in **one forward pass**. This increases activation memory (e.g., 887 MB to 1867 MB for K=2), but the peak memory remains **far below** backpropagation-based TTA (at least 6108MB) methods. As shown in Table 1 in the paper, multi-view inference improves accuracy with a moderate memory increase. If memory is too constrained to enable augmentation, PEA can still be run **without** extra views and already outperforms the backprop-free SOTA FOA, which requires 9 forward passes, while PEA needs only 2 forward passes with comparable memory usage.
>
> > **Comment 9:** For clarity, in the main text, please specify what a “block” corresponds to in the ViT architecture, and explicitly indicate that the referenced block 3 is the third out of nine blocks.
>
> Thank you for the suggestion. In our ViT experiments, a “block” refers to a **transformer encoder block**. The ViT model we use contains **12 encoder blocks in total**, but in Fig. 1 we visualize only **9 blocks** due to space and layout constraints. In this figure, “block 3” therefore corresponds to the **third encoder block** out of the 12 blocks. PEA applies alignment progressively at **every encoder block**, and we will clarify the block definition and the visualization selection in the final paper.
>
> > **Comment 10:** Please cite prior work using test-time augmentation (e.g., Simonyan and Zisserman, 2014).
>
> We have cited this paper in our revised version.

---

> ### Author Response · Authors · 2025-11-20
> **Response to Reviewer 9W85 (Part 3)**
>
> **​Summary and Request for Reconsideration:**
>
> We have substantially strengthened the paper by addressing your concerns: (1) clarifying that our novelty lies in explicit embedding alignment at test time ​**in contrast to** model updates adopted by conventional TTA methods; (2) adding source data ablation showing only ​**10% of source samples** are sufficient to obtain accurate statistics (Table R3-1); (3) providing ResNet visualizations demonstrating generality (Figures 7 and 8 in the revised paper); (4) citing GALA and PALM for layer-wise adaptation; (5) softening claims about geometric transformations; and (6) adding comprehensive new experiments (batch size ​of 1/2, mixed-domain settings).
>
> With ​**9 new tables, 3 new figures, and 2 new experimental settings​**, our method demonstrates: (1) **​strong TTA performance across 3 datasets** (CIFAR10-C, CIFAR100-C, ImageNet-C) **​on both CNN and ViT architectures** under ​**single-domain and mixed-domain​** settings, (2) successful edge deployment where all backprop methods fail (Table 4​ in the paper), and (3) a unique combination of backprop-free and architecture-agnostic design. If our responses adequately address your concerns, we would greatly appreciate reconsideration of the score. We remain available for additional clarifications during the discussion period.

---

> > ### Comment · Reviewer_9W85 · 2025-11-24
> >
> > I thank the authors for their responses and additional experiments. However, at this point, my decision remains unchanged for the following reasons:
> >
> > -   **First contribution is not novel, and some claims are insufficiently supported.**
> >
> >     -   The statement _“Our analysis of intermediate embeddings uncovers the essence of domain shifts, which can be characterized as translations, scalings, and rotations of the embedding space”_ is presented as a novel insight, but the underlying idea—that distribution shifts modify the statistical descriptors of the dataset—is a well-established fact. This is foundational to density-based OOD detection methods (e.g., Shafaei et al. [1], Yang et al. [5]), which address a more general setting than what is claimed here as the “essence of distribution shifts.” Overall, prior work in this space is insufficiently acknowledged and carefully reviewed (e.g., Rabanser et al. [2]).
> >
> >     -   The idea that the mean and covariance of the distribution shift across layers is also not new. Lee et al. [3] and Quintanilha et al. [4] show that distribution shifts can be detected from layerwise changes in hidden-representation statistics.
> >
> >     -   The claim that mean shifts “resemble” rotation or scaling cannot be made solely on the basis of visualization. These transformations are linear operators and should be supported by some empirical evidence showing that such operators approximate the observed shifts (e.g., Albalak et al.).
> >
> > -   **Source-dependence limitation.**
> >     The claim that using only 10% of the source data is sufficient to estimate accurate statistics and yields performance comparable to using 100%, or that the impact of this fraction may be neglected, is not supported. A rigorous validation is necessary. Relevant insights can be found in Sorscher et al. regarding scaling laws under data pruning, as well as broader work on data selection [6].
> >
> > -   **Scope of domain shifts.**
> >     While the paper is positioned as addressing domain shifts, it only considers synthetic image corruptions. Evaluating the method on additional domain-shift benchmarks—especially those from DomainBed—is crucial to assessing its generality and validity (e.g., VLCS, PACS, OfficeHome, TerraIncognita). See Gulrajani et al. [8] and Iwasawa et al. [9]. Could the authors clarify why this was not addressed or discussed?
> >
> >
> > **References:**
> >
> > -   [1] Shafaei, A., Schmidt, M., & Little, J. “A less biased evaluation of out-of-distribution sample detectors.” (2018).
> >
> > -   [2] Rabanser, S., Günnemann, S., & Lipton, Z. “Failing loudly: An empirical study of methods for detecting dataset shift.” NeurIPS (2019).
> >
> > -   [3] Lee, K., et al. “A simple unified framework for detecting out-of-distribution samples and adversarial attacks.” NeurIPS (2018).
> >
> > -   [4] Quintanilha, I. M., et al. “Detecting out-of-distribution samples using low-order deep features statistics.” (2019).
> >
> > -   [5] Yang, J., et al. “Generalized out-of-distribution detection: A survey.” IJCV (2024).
> >
> > -   [6] Sorscher, B., et al. “Beyond neural scaling laws: beating power law scaling via data pruning.” NeurIPS (2022).
> >
> > -   [7] Albalak, A., et al. “A survey on data selection for language models.” (2024).
> >
> > -   [8] Gulrajani, I., & Lopez-Paz, D. “In search of lost domain generalization.” (2020).
> >
> > -   [9] Iwasawa, Y., & Matsuo, Y. “Test-time classifier adjustment module for model-agnostic domain generalization.” NeurIPS (2021).

---

> > > ### Author Response · Authors · 2025-11-26
> > > **Follow-up Response to Reviewer 9W85 (Part 2)**
> > >
> > > > Scope of domain shifts.
> > >
> > > **Response**: Our paper is positioned in the **online test-time adaptation (OTTA)** setting, where the benchmarks in prior works are common datasets including **CIFAR10-C, CIFAR100-C, and ImageNetC**, which provide 15 domains. Many existing OTTA papers are evaluated solely on these benchmarks including Tent \[1\], EATA \[2\], EcoTTA \[3\], SAR \[4\], MECTA \[5\], and L-TTA \[6\]. As summarized in the recent survey \[7\], these 3 datasets have been considered the **standard OTTA benchmarks**.
> > >
> > > The datasets in DomainBed are primarily used in domain generalization, but we agree that more evaluation on them can help assess the generalizability.  Due to the rebuttal time limitation, we **add** an additional evaluation on **PACS** dataset. Specifically, we use the photo domain as the source training set and use art painting, cartoon and sketch as the TTA domains. We report the average TTA accuracy using 2 models in Table R3-C. The results show that **our method PEA also performs effectively on the PACS dataset**.
> > >
> > > Table R3-C. TTA accuracy (%) on PACS dataset.
> > >
> > > | Model | Method |  |  |  |  |  |  |  |
> > > | :---: | :---: | :---: | :---: | :---: | :---: | :---: | :---: | ----- |
> > > | ViT-tiny | No Adapt | SAR | Tent | EATA | FOA (F \= 9\) | CMF | SPA | PEA |
> > > |  | 25.6 | 29.8 | 29.9 | 29.9 | 35.5 | 33.1 | 19.3 | 36.0 |
> > > | ResNet-50 | No Adapt | Tent | EATA | MECTA | L-TTA | CMF | PEA |  |
> > > |  | 30.7 | 23.9 | 23.9 | 45.4 | 23.7 | 23.5 | 47.8 |  |
> > >
> > > \[1\] Wang D, Shelhamer E, Liu S, et al. Tent: Fully test-time adaptation by entropy minimization. ICLR’21.
> > > \[2\] Niu S, Wu J, Zhang Y, et al. Efficient test-time model adaptation without forgetting, ICML’22.
> > > \[3\] Song J, Lee J, Kweon I S, et al. Ecotta: Memory-efficient continual test-time adaptation via self-distilled regularization. CVPR’23.
> > > \[4\] Niu S, Wu J, Zhang Y, et al. Towards stable test-time adaptation in dynamic wild world. ICLR’23.
> > > \[5\] Hong J, Lyu L, Zhou J, et al. Mecta: Memory-economic continual test-time model adaptation. ICLR’23.
> > > \[6\] Shin J, Kim H. L-TTA: Lightweight Test-Time Adaptation Using a Versatile Stem Layer. NeurIPS’24
> > > \[7\] Liang J, He R, Tan T. A comprehensive survey on test-time adaptation under distribution shifts.. International Journal of Computer Vision, 2025

---

> > > > ### Comment · Reviewer_9W85 · 2025-11-28
> > > >
> > > > I thank the authors for the clarifications and the additional results.
> > > >
> > > > * The reconstruction experiment provides stronger evidence than the visualizations in showing that the observed shifts can, to some extent, be approximated by linear operations. Taken together, these experiments are sufficient as supporting evidence.
> > > >
> > > >
> > > > * A rigorous assessment of the claimed **"performance saturation”** should include analyses on more than CIFAR100-C to rule out conclusions driven by the dataset-specific artifacts.
> > > >
> > > >
> > > > * The survey you cite does not include layer-selection methods, which makes it incomplete for accurately situating this work within the existing literature. It is important that new methods compare against prior approaches in order to properly assess progress in the field. Furthermore, I do not agree that evaluating solely on synthetic corruption datasets is sufficient to assess your method's performance. Some unsupervised domain adaptation (UDA) datasets such as DomainNet[1] or DomainBed (which is well adequate for UDA as well) should be evaluated equivalently.
> > > >
> > > > [1] Peng, Xingchao, et al. "Moment matching for multi-source domain adaptation." Proceedings of the IEEE/CVF international conference on computer vision. 2019.

---

> > > > > ### Author Response · Authors · 2025-12-01
> > > > > **Follow-up Response to Reviewer 9W85**
> > > > >
> > > > > > A rigorous assessment of the claimed **"performance saturation”** should include analyses on more than CIFAR100-C to rule out conclusions driven by the dataset-specific artifacts.
> > > > >
> > > > > **Response**: We thank the reviewer for the comments. Here we add a more detailed experiment to show the impact of source data usage for statistics calculation to the final TTA performance. Specifically, we use ViT and  ResNet50 architectures on CIFAR10-C, CIFAR100-C and PACS datasets to show that this effect is not dataset-specific. The results show that using only a small subset of the training data is sufficient to achieve nearly the same adaptation performance as using the full source dataset.
> > > > >
> > > > > Table R3-D. Impact of the fraction of source data used to adaptation accuracy(%).
> > > > >
> > > > > | Source Data Used | CIFAR10-C ViT | CIFAR100-C ViT | PACS ResNet-50 |
> > > > > | :---: | ----- | ----- | :---: |
> > > > > | 5% | 83.2±0.1 | 76.4±0.2 | 46.9±0.1 |
> > > > > | 10% | 83.2±0.1 | 76.8±0.2 | 47.2±0.1 |
> > > > > | 20% | 83.4±0.1 | 77.0±0.1 | 47.9±0.1 |
> > > > > | 50% | 83.4±0.1 | 77.0±0.1 | 48.1±0.1 |
> > > > > | 100% | 83.4±0.1 | 77.0±0.1 | 47.9±0.1 |
> > > > >
> > > > > > The survey you cite does not include layer-selection methods, which makes it incomplete for accurately situating this work within the existing literature. It is important that new methods compare against prior approaches in order to properly assess progress in the field.
> > > > >
> > > > > **Response**: We noticed a few layer-selection TTA methods \[1, 2\]. Park J et al. \[1\] proposes LAW, a weighted layer-update strategy that assigns different importance to layers during adaptation. In response to your suggestion, We **add LAW\[1\] as an additional baseline** and evaluate it on ResNet-50 across two datasets. From Table R3-E, it is clear that PEA achieves better performance, while maintaining backprop–free.  We also consider GALA \[2\] to be related to our approach, but its code is not yet publicly available. We will add \[2\] as a baseline in the camera-ready version if the implementation is released in time.
> > > > >
> > > > > Table R3-E. Adaptation accuracy(%) on CIFAR10-C and CIFAR100-C (LAW is the new baseline method).
> > > > >
> > > > > | Dataset | No Adapt | TENT | EATA | MECTA | EcoTTA | L-TTA | CMF | LAW | PEA |
> > > > > | :---: | :---: | :---: | :---: | :---: | :---: | :---: | :---: | :---: | :---: |
> > > > > | CIFAR10-C | 62.5 | 81.2 | 81.3 | 82.0 | 80.2 | 81.2 | 78.6 | **81.8** | **83.4** |
> > > > > | CIFAR100-C | 33.5 | 49.2 | 50.0 | 49.7 | 45.6 | 50.9 | 48.8 | **51.0** | **54.6** |
> > > > >
> > > > > \[1\] Park J et al. Layer-wise auto-weighting for non-stationary test-time adaptation, WACV’24.
> > > > > \[2\] Sahoo, Sabyasachi, et al. A layer selection approach to test time adaptation.AAAI’25.
> > > > >
> > > > > > Furthermore, I do not agree that evaluating solely on synthetic corruption datasets is sufficient to assess your method's performance. Some unsupervised domain adaptation (UDA) datasets such as DomainNet\[1\] or DomainBed (which is well adequate for UDA as well) should be evaluated equivalently.
> > > > >
> > > > > **Response**: Following your suggestion, we have added an evaluation on the UDA dataset PACS in Table R3-C. To better support our method, we further add the evaluation on the DomainNet. Specifically, we pretrain on the **Real** domain and then adapt continuously to the five target domains (Clipart, Infograph, Painting, Quickdraw, Sketch), reporting the average accuracy across them. As shown in Table R3-F, the results on DomainNet show that PEA remains competitive and effective on **realistic UDA benchmarks**, not only on synthetic corruption datasets.
> > > > >
> > > > > Table R3-F. Adaptation accuracy(%) on DomainNet.
> > > > > | Model |  |  |  | Method |  |  |  |
> > > > > | :---: | :---: | :---: | :---: | :---: | :---: | :---: | :---: |
> > > > > | ViT-Base | SAR | Tent | EATA | FOA (F \= 9\) | CMF | SPA | **PEA** |
> > > > > |  | 26.2 | 27.8 | 26.2 | **28.3** | 25.2 | 23.0 | **28.3** |
> > > > > | ResNet-50 | TENT | EATA | MECTA | L-TTA | CMF | LAW | **PEA** |
> > > > > |  | 30.3 | 30.4 | 31.1 | 30.7 | 32.1 | 30.6 | **33.2** |
> > > > >
> > > > > **Summary**: Through our rebuttal and additional response, we have provided 13 new tables and 3 new figures. Compared to prior TTA works, which typically evaluate on only a few datasets with a limited set of baselines, our study now covers 5 datasets and 11 baseline methods (including 2 additional datasets and 2 new baselines added in the rebuttal), substantially strengthening our contribution through a much more comprehensive evaluation across diverse scenarios. Furthermore, we strongly believe that our method's unique combination of backprop-free, architecture-agnostic, and edge-deployable properties represents a significant contribution to OTTA.

---

> ### Author Response · Authors · 2025-11-26
> **Follow-up Response to Reviewer 9W85 (Part 1)**
>
> > First contribution is not novel.
>
> **Response:** We thank the reviewer’s comment and would like to add further clarifications.
>
> First, we clarify that rather than merely noting that low-order statistics of intermediate features change under domain shift (as commonly done in OOD works), our analysis goes deeper: we systematically examine the embedding geometry and summarize the shift mainly as translations, scalings, and rotations from a visualization perspective. This deeper analysis serves as the basis of our method, providing a principled foundation for TTA method design.
>
> Regarding novelty compared to OOD works, we would like to clarify that PEA is essentially an **online test-time adaptation (OTTA)** problem [1]. It aims to align the source model with a shifted distribution online, instead of detecting OOD samples. Moreover, although OOD works use the intermediate statistics for outlier detection, they stop after the detection step, while OTTA will further adapt the model.
>
> Finally, **our core contribution is the design of a simple yet effective OTTA method based on the statistical changes of layer-wise embeddings. Compared to existing OTTA methods, PEA is backprop-free and applies to both CNN and transformer-based architectures (i.e., architecture-agnostic). We believe these two merits can mark PEA as a big advancement to the OTTA field.**
>
> [1] Liang J, He R, Tan T. A comprehensive survey on test-time adaptation under distribution shifts. IJCV, 2025.
>
> > The claim that mean shifts “resemble” rotation or scaling cannot be made solely on the basis of visualization.
>
> **Response:** Our visualization results across multiple layers and domains(Figs. 2–9 in the Appendix) consistently show systematic changes of intermediate embeddings. In particular, Fig. 6 in Appendix compares the features before and after PEA: the shifted features are visibly corrected to a large extent by layer-by-layer alignment and the final features move much closer to the source.
>
> To better support the claim, we add **a new quantitative experiment**. While it is hard to directly prove the feature shift can be decomposed into linear transformations (translation, scaling, and rotation), we instead experiment whether **such linear operations can reconstruct the shifted features back toward the source with low error**. If this reconstruction is effective, it suggests that domain shift induces intermediate embedding changes that include non-trivial linear transformation components. Specifically, we extract the shifted and source features from CIFAR10-C  for each block on ViT and apply 3 linear transformations including **mean alignment, mean + variance alignment, and WCT** (additionally aligns the covariance) and measure the relative Frobenius error and R² between the transformed features and source features, averaged across 15 domains as shown in Table R3-A.
>
> We can see that the transformed features show clearly lower error and higher R² compared to the raw shifted features (Err\_raw and R²\_raw), indicating that **a substantial portion of the domain shift in intermediate embeddings can be explained and mitigated by linear statistic-based transformations**. While we do not claim the shift is purely linear, this provides concrete evidence that such transformations are **beneficial for TTA**, especially when combined with our progressive alignment in our PEA.
>
> Table R3-A. Reconstruction error and R² on CIFAR10-C using linear transformations.
> | ViT Layer (Block) | Err\_raw | Err\_mean | Err\_var | Err\_WCT | R² \_raw | R² \_mean | R² \_var | R² \_WCT |
> | :---: | :---: | :---: | :---: | :---: | :---: | :---: | :---: | :---: |
> | Block 1 | 0.02 | 0.01 | 0.01 | 0.01 | 0.55 | 0.85 | 0.88 | 0.89 |
> | Block 3 | 0.10 | 0.05 | 0.05 | 0.04 | \-0.47 | 0.73 | 0.72 | 0.75 |
> | Block 5 | 0.21 | 0.11 | 0.11 | 0.10 | \-0.66 | 0.59 | 0.55 | 0.62 |
> | Block 7 | 0.22 | 0.13 | 0.14 | 0.13 | \-0.37 | 0.56 | 0.51 | 0.55 |
> | Block 9 | 0.22 | 0.15 | 0.16 | 0.15 | 0.06 | 0.54 | 0.47 | 0.55 |
> | Block 11 | 0.30 | 0.27 | 0.29 | 0.27 | 0.45 | 0.58 | 0.51 | 0.58 |
>
> > Source-dependence limitation.
>
> **Response:** In the rebuttal, we randomly subsampled the source data 5 times for each fraction. We add the mean with std on CIFAR100-C in the Table R3-B. These results show that performance **saturates quickly**. We also note the scaling-law analysis in \[6\]. It is crucial for [6] because the dataset size directly affects the model training process and the performance is highly sensitive to the amount of training data. In our case, the source data are used **only once** offline to estimate the statistics. Our ablation suggests that, in practice, source dependence is modest rather than a critical limitation. We will add a more comprehensive validation in the final paper.
>
> Table R3-B. Impact of source data used to estimate statistics on CIFAR100-C.
> | Source Data Used | Accuracy (%) |
> | :---: | :---: |
> | 10% | 76.8±0.2 |
> | 20% | 77.0±0.1 |
> | 100% | 77.0±0.1 |

---

### Official Review · Reviewer_bwXx · 2025-10-31

**Soundness:** 3
**Presentation:** 3
**Contribution:** 3
**Rating:** 6
**Confidence:** 4

**Summary:**

This paper proposes Progressive Embedding Alignment (PEA), a backpropagation-free and architecture-agnostic Test-Time Adaptation (TTA) method. The key idea is to interpret domain shifts as three geometric transformations in the embedding space—translation (mean shift), scaling (variance shift), and rotation (covariance shift)—and correct them progressively via Whitening-Coloring Transform (WCT)–based covariance alignment. PEA performs only two forward passes per test batch, without modifying model weights, and can be applied to both CNNs and ViTs. Experiments on CIFAR10-C/100-C and ImageNet-C show that PEA achieves competitive accuracy with minimal memory and latency, demonstrating its suitability for real-time or edge deployment.

**Strengths:**

The embedding-space interpretation of domain shift (translation, scaling, rotation) is elegant and intuitive. It provides a clear geometric motivation for the proposed covariance alignment strategy.

PEA is architecture-agnostic for both CNNs and ViTs.

The fully forward-only, memory-light, and backprop-free natures make PEA ideal for edge devices. It achieves low latency (~0.3s) and small memory footprint (<1GB) while maintaining strong accuracy.

Device-level tests on Jetson Orin Nano enhance practical relevance.

**Weaknesses:**

As the proposed method relies on statistical estimation, its performance degrades under small batch sizes. And, I am wondering how does the method performs when the batch size is as small as 2? Moreover, with augmentation enabled, could PEA still operate effectively with a batch size of 1?

**Questions:**

How to decide the threshold for detecting domain changes (Eq. 7)  across different models and domains during online testing? An ablation study (with and without Eq.7) and sensitivity analysis on this threshold would be helpful.

How does the number of augmented views in PEA affect performance? More detailed sensitivity analyses on this factor are preferred.

How does PEA perform under mixed domain shifts? It would be helpful to see results for both CNN and ViT backbones in such scenarios.

The reported results of SPA appear to be lower than those in the original paper. Could the authors clarify this discrepancy? Since SPA is a strong backpropagation-based method, it is understandable that PEA (focus on forward-only) has lower performance than SPA, but ensuring consistent and accurate SPA results would make the comparison more convincing.

Suggestion:

FOA also includes an alignment component, namely “activation shifting,” which appears conceptually related to the proposed PEA. However, PEA performs alignment in a more fine-grained, layer-wise, and comprehensive manner. It would be much better to discuss this relationship in the paper.

I would like to consider increasing my score if these are well addressed.

---

> ### Author Response · Authors · 2025-11-20
> **Response to Reviewer bwXx （Part 1）**
>
> > **Comment 1:** As the proposed method relies on statistical estimation, its performance degrades under small batch sizes. And, I am wondering how does the method performs when the batch size is as small as 2? Moreover, with augmentation enabled, could PEA still operate effectively with a batch size of 1?
>
> We appreciate this insightful comment, and it helps us improve the paper\! Initially, we also thought that accurate covariance estimation is very difficult with tiny batches, therefore only ran experiments with BS down to 4\. However, during rebuttal, we conducted new experiments with BS=1 and BS=2. As shown in Table R2-1, PEA still outperforms other baselines in BS=1 and performs robustly in BS \=2. After a thorough analysis, we conjecture that the good performance is **attributed to the EMA-based statistic update**. Specifically, PEA is designed to work in a streaming regime, where an Exponential Moving Average (EMA) strategy is employed to estimate statistics by accumulating across historical batches.
> To validate this conjecture, we further run another ablation by removing EMA for different batch sizes. As shown in Table R2-2, EMA is crucial, especially for small batch sizes, because per-batch statistics from very few samples are too noisy for alignment. This indicates EMA effectively aggregates information over the stream, stabilizing the target statistics and making the alignment reliable even when each individual batch contains only one sample. We will add the new results to the final paper.
>
> Table R2-1. Results on BS=1 and BS=2 on CIFAR100-C and ImageNet-C with ViT-Base.(“F” indicates that the method fails under BS=1.)
>
> | Method | ImageNet-C |  | CIFAR100-C |  |
> | :---: | :---: | :---: | :---: | :---: |
> |  | Batch Size \= 1 | Batch Size \= 2 | Batch Size \= 1 | Batch Size \= 2 |
> | No Adapt | 55.5 | 55.5 | 61.6 | 61.6 |
> | SAR | 58.6 | 62.0 | 62.0 | 64.2 |
> | CMF | F | 61.5 | F | 61.5 |
> | FOA (K=9) | F | 64.4 | F | 66.9 |
> | SPA | 59.5 | 60.8 | 46.5 | 70.6 |
> | **PEA** | 61.6 | 62.5 | 69.5 | 69.9 |
>
> Table R2-2. Effect of EMA under different batch sizes on CIFAR100-C
>
> | Batch size | 1 | 2 | 4 | 16 | 64 |
> | :---: | :---: | :---: | :---: | :---: | :---: |
> | PEA w/o EMA | 0.5 | 1.8 | 6.9 | 53.5 | 68.3 |
> | PEA with EMA | 69.5 | 69.9 | 70.0 | 75.7 | 77.0 |
>
> > **Comment 2:** How to decide the threshold for detecting domain changes (Eq. 7\) across different models and domains during online testing? An ablation study (with and without Eq.7) and sensitivity analysis on this threshold would be helpful.
>
> We use the prediction entropy for spike detection because it is a simple and direct indicator of prediction uncertainty. When a domain shift occurs, this uncertainty typically **spikes** due to the mismatch between source-trained decision boundaries and shifted inputs. The threshold is set to 1 in the paper based on our empirical analysis. Specifically, we analyzed the prediction entropy across batches and found that the entropy usually ranges from 0 to 0.8 in the training set. Therefore, we empirically set the threshold as 1\. To evaluate the impact of the threshold selection, we added additional experiments with different threshold values during the rebuttal. As shown in Table R2-3, the impact on accuracy is marginal. This is because (1) shift detection is mainly valid at the boundary of the domain transitions and (2) as long as the threshold can differentiate two domains, the performance will not be affected.
>
> Table R2-3. Impact of the shift detection threshold on accuracy (%).
>
> | Dataset | 0.5 | 1.0 | 1.5 | 2.0 |
> | :---: | :---: | :---: | :---: | :---: |
> | ImageNet-C | 66.4 | 66.5 | 66.4 | 66.4 |
> | CIFAR100-C | 77.0 | 77.0 | 77.1 | 76.8 |
>
> > **Comment 3:** How does the number of augmented views in PEA affect performance? More detailed sensitivity analyses on this factor are preferred.
>
> We thank the reviewer for the suggestion.  Our default setting in the paper is K=2 and during rebuttal, we added another ablation on the number of augmentation views from K=0 to 4 in the following table R2-4. We observe that multi-view augmentation consistently improves performance over K=0 (no augmentation), with the largest gain from K=0 to K=1, and a smaller but still clear gain from K=1 to  K=2. Beyond K=2, the improvements saturate and become marginal (\<0.1%). Since increasing K also increases compute and memory, we adopt **K=2** as the default to capture most of the benefit of augmentation while keeping the overhead modest. We will add the results to the final manuscript.
>
> Table R2-4. Impact of the number of test-time augmentation views K on top-1 accuracy (%).
>
> | Dataset | K \= 0 | K \= 1 | K \= 2 | K \= 3 | K \= 4 |
> | :---: | :---: | :---: | :---: | :---: | :---: |
> | ImageNet-C | 64.5 | 66.2 | 66.5 | 66.6 | 66.6 |
> | CIFAR100-C | 75.7 | 76.6 | 77.0 | 77.0 | 77.1 |

---

> ### Author Response · Authors · 2025-11-20
> **Response to Reviewer bwXx （Part 2）**
>
> > **Comment 4:** How does PEA perform under mixed domain shifts? It would be helpful to see results for both CNN and ViT backbones in such scenarios.
>
> Thanks for this insightful comment\! Our main experiments follow the standard continual single-domain protocol used in prior CTTA works, but we do agree that the mixed-domain batch is a more challenging case and worth exploring.
>
> First, we constructed a mixed CIFAR100-C dataset where all 15 corruptions (severity 5\) are combined into a single pool and randomly shuffled, i.e., each batch contains samples from multiple domains. This ensures that each batch of 64 samples contains 4.3 samples on average per corruption type. This represents a realistic scenario where the model encounters rapid, unpredictable domain shifts within individual batches, which is more challenging than the single-domain setting where each batch consists of samples from a single corruption type.
>
> As shown in Table R2-5, PEA with ViT achieves **72.0%**, a substantial improvement over the **61.6%** source model, and outperforms all the baselines.
>
> Table R2-5. Results on mixed-domain data on CIFAR100-C with ViT-Base.
>
> | Dataset | Source Model | Tent | EATA | SAR | FOA (K=9) | CMF | SPA | PEA |
> | :---: | :---: | :---: | :---: | :---: | :---: | :---: | :---: | :---: |
> | CIFAR100-C | 61.6 | 61.2 | 61.2 | 62.0 | 62.6 | 69.0 | 71.4 | 72.0 |
>
> To understand why PEA remains effective, we visualize the intermediate embeddings for 9 ViT blocks using t-SNE, comparing **clean data vs. mixed-domain data** (please see Appendix G and Fig. 9 in the revised paper). As expected, mixed-domain batches exhibit **lower intra-class compactness** because samples originate from heterogeneous domains. However, the key observation is that the **structural distortion is still systematic** across blocks: class clusters undergo consistent **mean shift**, **variance shift**, and **channel-wise covariance shift**.
>
> Since PEA aligns features via a WCT and applies **layer-wise distance-aware weighting** together with **EMA smoothing**, it corrects these dominant geometric shifts even when the batch distribution is a mixture domain. This explains why PEA continues to improve accuracy significantly under mixed-domain TTA.
>
> Additionally, we want to clarify that ‘mixed-domain’ actually refers to the semantic characteristics defined by humans (e.g., fog, snow). However, at the embedding space, there might be another high-dimensional space where these mixed domains exhibit similar features. As a result, TTA approaches are still able to find a proper perspective for adaptation or alignment.
>
>
> > **Comment 5:** The reported results of SPA appear to be lower than those in the original paper. Could the authors clarify this discrepancy? Since SPA is a strong backpropagation-based method, it is understandable that PEA (focus on forward-only) has lower performance than SPA, but ensuring consistent and accurate SPA results would make the comparison more convincing.
>
> Thank you for pointing this out. The gap between our SPA numbers and those reported in Niu et al. (2025) comes from a difference in the evaluation protocol rather than from an implementation issue. In our main experiments (Table 1 in the paper), all methods — including SPA and PEA — are evaluated under the *lifelong continual* TTA setting (typical TTA setting), where the 15 ImageNet-C domains are streamed sequentially and the adapted model is **not reset** between domains. In contrast, the SPA paper reports results under an *independent per-domain* protocol, where the model is reset to the source checkpoint before adapting to each corruption type.
> Using the authors’ official SPA implementation and hyperparameters, we observe that SPA achieves higher accuracy under this per-domain-reset protocol, consistent with the original paper, and in this case PEA does **not** outperform SPA, which is reasonable given that SPA performs both data augmentation and backpropagation. We have added a note in Section 5 (Page 7\) of the revised manuscript to clarify this discrepancy.

---

> ### Author Response · Authors · 2025-11-20
> **Response to Reviewer bwXx （Part 3）**
>
> > **Comment 6:** FOA also includes an alignment component, namely “activation shifting,” which appears conceptually related to the proposed PEA. However, PEA performs alignment in a more fine-grained, layer-wise, and comprehensive manner. It would be much better to discuss this relationship in the paper.
>
> We appreciate the reviewer’s suggestion and agree that FOA’s **“activation shifting”** is conceptually related to our alignment strategy. However, the mechanisms and scope are quite different. In FOA, activation shifting is applied only to the **final-layer CLS token**, modeling a simple shift of the feature center and is used as an **auxiliary regularization term**. As reported in FOA’s ablation, this component brings only a modest gain (≈0.9%); the main improvement comes from their **evolutionary strategy with many forward passes** to optimize a prompt, which is shown primarily for transformer-style models and is not effective for CNN architectures.
>
> In contrast, PEA starts from the empirical observation that domain shift induces a **systematic embedding-space shift across layers** (in mean, variance, and covariance, while FOA only identifies mean) for both ViT and ResNet. Our method is **statistics-driven**: it applies covariance alignment block by block, explicitly pulling target embeddings back toward the source distribution. This yields a **fine-grained, progressive layer-wise alignment** of the entire representation, rather than a single shallow shift of the last layer. Empirically, PEA achieves **higher accuracy with fewer forward passes** than FOA (e.g., In Table R2-1, PEA achieves 69.5% while FOA fails at batch size of 1; In Table 2 in the paper, PEA achieves 77.0% vs. FOA’s 68.0% at batch size of 64 on CIFAR100-C), while remaining fully backprop-free and architecture-agnostic.
>
> Another key distinction in methodology is that FOA keeps the backbone weights frozen but learns a small set (e.g., 3\) of input prompts in the first layer. These prompts are optimized at test time, thereby fitting the model to the new domain, which requires multiple forward passes (up to 27). In contrast, PEA keeps the pre-trained model completely intact, and aligns the intermediate embeddings to fit the source model, **achieving a completely backprop-free and model-agnostic approach with only 2 forward passes.** We will clarify this relationship and distinction in the final paper.

---

### Official Review · Reviewer_mDae · 2025-10-31

**Soundness:** 4
**Presentation:** 4
**Contribution:** 4
**Rating:** 10
**Confidence:** 5

**Summary:**

This paper proposes **Progressive Embedding Alignment (PEA)**, a novel Test-Time Adaptation (TTA) method that is **backpropagation-free**, **architecture-agnostic**, and highly efficient. The authors aim to overcome two major limitations of existing TTA work: 1) the high computational and memory overhead of backpropagation-based methods (e.g., Tent, EATA), which makes them unsuitable for edge devices , and 2) the architecture-specificity of recent efficient methods, which are often tailored for either CNNs or ViTs, but not both.

The core idea is motivated by a principled analysis of domain shift. The authors empirically demonstrate that domain shifts induce a combination of three consistent geometric distortions in the intermediate embedding space:
1.  **Translation** (mean shift)
2.  **Scaling** (variance shift)
3.  **Rotation** (channel-wise covariance shift)

PEA works by explicitly correcting these distortions. In an offline step, it pre-computes and stores the source-domain statistics (mean $\mu_{s,l}$ and covariance $\Sigma_{s,l}$) for each block $l$. Then, at test time, it uses a two-pass, forward-only procedure:
* **Pass 1:** Estimates the statistical distance between the current batch and the source to compute a layer-wise alignment weight $w_l$.
* **Pass 2:** Applies a Whitening-Coloring Transform (WCT) to realign the batch features to the source statistics. The final, corrected feature $F_l'$ is a weighted blend of the original and aligned features, $F_{l}^{\prime}=(1-w_{l})F_{l}+w_{l}Y_{l}$, which prevents over-correction .

**Strengths:**

- Clear Motivation and Novelty: The analysis of domain shift as a geometric transformation (translation, scaling, rotation) in the embedding space is intuitive and provides a strong foundation for the proposed method. The idea of reversing these distortions via a weighted WCT, without any gradient updates, is a novel and elegant approach to TTA.

- Exceptional Efficiency (Memory and Latency): The paper's primary contribution is its practicality. By being backprop-free and requiring only two forward passes , PEA is significantly more efficient than SOTA methods like CMF, SPA, or EATA. This is powerfully demonstrated in the edge device evaluation (Table 4), where PEA runs successfully on a Jetson Orin Nano, while numerous backprop-based baselines (Tent, EATA, CMF, etc.) fail due to OOM errors.

- Strong Empirical Results: PEA (especially with augmentation) achieves state-of-the-art or highly competitive accuracy on all three benchmarks (ImageNet-C, CIFAR10-C, CIFAR100-C).

- Generality (Architecture-Agnostic): This is a key advantage. The method works "out-of-the-box" for both CNNs (ResNet-50) and Transformers (ViT-Base) using the identical procedure

**Weaknesses:**

- Source Data Requirement: The method's primary limitation, which is acknowledged by the authors26, is that it is not "source-free." It requires access to the original source training dataset in the offline stage to pre-compute the source statistics ($\mu_{s,l}$, $\Sigma_{s,l}$)27. While this is a valid setup for TTA, it is a stronger assumption than methods that only require the pre-trained model.

-  Storage Overhead: While the inference compute/memory is low, PEA introduces a storage cost by requiring the mean vector and covariance matrix for each aligned block to be stored. The paper mentions this is minimal (~30MB for ViT-Base), but this cost should be explicitly reported for all models in the main tables to provide a complete picture of the resource trade-offs.

- Statistics Estimation: The method's effectiveness hinges on the quality of the estimated test-time statistics ($\mu_{t,l}$, $\Sigma_{t,l}$)28. While the EMA helps, the authors note that performance could degrade with extremely small batches (e.g., BS=1) or high class imbalance29. A formal evaluation at BS=1, a true "online" setting, would be valuable to fully test this boundary condition.

**Questions:**

- Are recent methods used to reduce the overhead of backprop only tied to vit or cnn or have high latency? if so can you share related work to that?

-  In equation 1 why you didn't use the mahalanobis distance between each input in the batch and the source distribution instead of relying on the batch statistics and in that case even for bs 1 the method shall work preoperly out of the box without having to think about the scale difference between mean and variance as the current distance can be overwhelmed by one measure over the other (if we have more mean shift or more variance).

- Can you ablate the choice of distance and what happens if we simply use the shift in mean ?

- how do you select the fixed threshold used to flag spikes in feature alignment

- Can you ablate the choice of k views augmentaion to see its effect on the main results ?

---

> ### Author Response · Authors · 2025-11-20
> **Response to Reviewer mDae (Part 1)**
>
> > **Comment 1:** Source Data Requirement: The method's primary limitation, which is acknowledged by the authors, is that it is not "source-free." It requires access to the original source training dataset in the offline stage to pre-compute the source statistics. While this is a valid setup for TTA, it is a stronger assumption than methods that only require the pre-trained model.
>
> we agree with the reviewer and explicitly acknowledged this limitation in the original submission (Line 484-485). There are indeed some methods (like Tent, SAR) that are completely source-free. However, many recent TTA works (e.g.,  EcoTTA, EATA, FOA, L-TTA) that yield better performance and efficiency also require source data for one-time offline pre-computing, similar to our work PEA.
>
> To alleviate this issue, we conducted an additional experiment to evaluate the impact of source data size. The result indicates that as few as 10% of source samples are sufficient to obtain accurate statistics for CIFAR100 as the Table R1-1 shows. Moreover, this preprocessing is performed **once and entirely offline** before deployment; at test time, PEA only uses the stored statistics and never accesses source data again. We will add the discussion of the impact of source data size in the camera-ready manuscript.
>
> Table R1-1. Impact of the fraction of source data used to estimate statistics on CIFAR100-C performance with PEA.
>
> | Source Data Used | Accuracy (%) |
> | :---: | :---: |
> | 10% | 76.8 |
> | 20% | 77.0 |
> | 100% | 77.0 |
>
> > **Comment 2:** Storage Overhead: While the inference compute/memory is low, PEA introduces a storage cost by requiring the mean vector and covariance matrix for each aligned block to be stored. The paper mentions this is minimal (\~30MB for ViT-Base), but this cost should be explicitly reported for all models in the main tables to provide a complete picture of the resource trade-offs.
>
> We appreciate this comment and will add the storage cost to the main table in the final version. PEA stores per-block source means and covariances as lightweight metadata computed once offline. For ViT-Base, this adds **28 MB** on top of a **327 MB** model (≈**8.6%** overhead). For ResNet-50, we use grouped covariance square roots, which reduces the statistics to **3 MB** for a **90 MB** model (≈**3.3%** overhead). In all cases, this cost is fixed (independent of batch size and test-time duration) and is small compared to the model size.
>
> > **Comment 3:** Statistics Estimation: The method's effectiveness hinges on the quality of the estimated test-time statistics. While the EMA helps, the authors note that performance could degrade with extremely small batches (e.g., BS=1) or high class imbalance29. A formal evaluation at BS=1, a true "online" setting, would be valuable to fully test this boundary condition.
>
> We appreciate this insightful comment, and it helps us improve our work\! Initially, we also thought that accurate covariance estimation is very difficult with tiny batches, therefore only ran experiments with BS down to 4\. However, during rebuttal, we conducted new experiments with BS=1 and BS=2. As shown in Table R1-2, PEA still outperforms other baselines in BS=1 and performs robustly in BS \=2. After a thorough analysis, we conjecture that the good performance is **attributed to the EMA-based statistic update**. Specifically, PEA is designed to work in a streaming regime, where an Exponential Moving Average (EMA) strategy is employed to estimate statistics by accumulating across historical batches. To validate this conjecture, we further run another ablation by removing EMA for different batch sizes. As shown in Table R1-3, EMA is crucial, especially for small batch sizes, because per-batch statistics from very few samples are too noisy for alignment. This indicates EMA effectively aggregates information over the stream, stabilizing the target statistics and making the alignment reliable even when each individual batch contains only one sample. We will add the new results to the final paper.
>
> Table R1-2. Results on BS=1 and BS=2 on CIFAR100-C and ImageNet-C with ViT-Base.(“F” indicates that the method fails under BS=1.)
>
> | Method | ImageNet-C |  | CIFAR100-C |  |
> | :---: | :---: | :---: | :---: | :---: |
> |  | Batch Size \= 1 | Batch Size \= 2 | Batch Size \= 1 | Batch Size \= 2 |
> | No Adapt | 55.5 | 55.5 | 61.6 | 61.6 |
> | SAR | 58.6 | 62.0 | 62.0 | 64.2 |
> | CMF | F | 61.5 | F | 61.5 |
> | FOA (K=9) | F | 64.4 | F | 66.9 |
> | SPA | 59.5 | 60.8 | 46.5 | 70.6 |
> | **PEA** | 61.6 | 62.5 | 69.5 | 69.9 |
>
> Table R1-3. Effect of EMA under different batch sizes on CIFAR100-C
>
> | Batch size | 1 | 2 | 4 | 16 | 64 |
> | :---: | :---: | :---: | :---: | :---: | :---: |
> | PEA w/o EMA | 0.5 | 1.8 | 6.9 | 53.5 | 68.3 |
> | PEA with EMA | 69.5 | 69.9 | 70.0 | 75.7 | 77.0 |

---

> ### Author Response · Authors · 2025-11-20
> **Response to Reviewer mDae (Part 2)**
>
> > **Comment 4:** Are recent methods used to reduce the overhead of backprop only tied to vit or cnn or have high latency? if so can you share related work to that?
>
> Recent methods that aim to reduce backpropagation overhead in TTA generally fall into two categories. **(1) Reduced-backprop methods:** **EcoTTA** (CVPR’23) \[1\] and **L-TTA** (NeurIPS’24)\[2\] still rely on gradient updates but lower activation memory by updating only a subset of layers (e.g., meta layers or stem layers). These methods are designed for and evaluated primarily on **CNN** backbones with BatchNorm (e.g., ResNet). **(2) Backprop-free methods:** **FOA (ICML’24)** \[3\]**, LeanTTA (arXiv’25)** \[4\], and more recent Gaussian-alignment approaches such as **ADAPT (NeurIPS’25)** \[5\]. **FOA** optimizes prompts to adapt the final embeddings but requires **many** forward passes to achieve its best results (up to 27 passes). **LeanTTA** updates BN statistics by combining source and target statistics, but is limited to **CNN-based** models with BN layers. **ADAPT** is also backprop-free, but it is specifically designed for VLM (CLIP): it models class-conditional Gaussians in CLIP feature space and relies on a **text** encoder to obtain class-specific tokens, which **does not** match our single-modality setting. Instead, our PEA can work on both CNN and ViT models and is completely backprop-free. As highlighted in Line 308-317 of the original submission, another key perspective that distinguishes our work with others is: To handle domain shift, most prior works adapt the pre-trained model to fit the new data distribution, thereby requiring back-prop. While PEA aligns the distorted data distribution to fit the pre-trained source model, completely eliminating backprop.
>
> Reference:
> \[1\] Song J, Lee J, Kweon I S, et al. Ecotta: Memory-efficient continual test-time adaptation via self-distilled regularization\[C\]//Proceedings of the IEEE/CVF Conference on Computer Vision and Pattern Recognition. 2023: 11920-11929.
> \[2\] Shin J, Kim H. L-TTA: Lightweight Test-Time Adaptation Using a Versatile Stem Layer\[J\]. Advances in Neural Information Processing Systems, 2024, 37: 39325-39349.
> \[3\] Niu S, Miao C, Chen G, et al. Test-time model adaptation with only forward passes\[J\]. arXiv preprint arXiv:2404.01650, 2024\.
> \[4\] Dong C, Jia H, Kwon Y D, et al. LeanTTA: A Backpropagation-Free and Stateless Approach to Quantized Test-Time Adaptation on Edge Devices\[J\]. arXiv preprint arXiv:2503.15889, 2025\.
> \[5\] Zhang Y, Kim Y, Choi Y G, et al. Backpropagation-Free Test-Time Adaptation via Probabilistic Gaussian Alignment\[J\]. arXiv preprint arXiv:2508.15568, 2025\.
>
> > **Comment 5:** In equation 1 why you didn't use the mahalanobis distance between each input in the batch and the source distribution instead of relying on the batch statistics and in that case even for bs 1 the method shall work preoperly out of the box without having to think about the scale difference between mean and variance as the current distance can be overwhelmed by one measure over the other (if we have more mean shift or more variance).
>
> Our goal in Eq. (1) is to obtain a stable layer-wise shift weights for the next pass alignment. For this purpose, calculating the L2 distance between batch statistics (mean and variance) to the corresponding source statistics is simple, robust under small batches, and works well in practice. We use the Euclidean (L2) distance rather than Mahalanobis distance as a deliberate trade-off between robustness and efficiency: it avoids repeated covariance inversion which is more expensive, while still providing a monotonic measure of how much each block has drifted.
> To further validate our design choice, we added an additional ablation study comparing three distance metrics: **L1**, **L2**, and  **Mahalanobis distance**. The results on CIFAR100-C with ViT are shown below in Table R1-4. We can see that the L2 distance already performs strongly and is only **0.3%** below the Mahalanobis distance, which confirms that the simple L2 score is sufficient to capture the relative block-wise shift while avoiding the overhead of covariance-based distances. We also profiled latency on the edge device: the Mahalanobis variant adds measurable overhead, requiring 0.33 s per batch compared to 0.31 s for the L2 distance.
>
> Table R1-4. Impact of Pass-1 distance metric choice on CIFAR100-C using ViT-Base
>
> | Distance Metric | Acc (%) |
> | :---: | :---: |
> | L1 | 73.6 |
> | L2 (ours) | 77.0 |
> | Mahalanobis | 77.3 |

---

> ### Author Response · Authors · 2025-11-20
> **Response to Reviewer mDae (Part 3)**
>
> > **Comment 6:** Can you ablate the choice of distance and what happens if we simply use the shift in mean?
>
> We ablated the distance definition in Eq. (1) by comparing our default mean+variance distance to a variant that uses only the mean shift. As shown in Table R1-5, using only the mean significantly degrades accuracy. This suggests that **variance shift is also crucial** for measuring how strongly each block is affected by domain shift. We will add this ablation to the final paper.
>
> Table R1-5. Impact of distance choice on ImageNet-C using ViT-Base.
>
> | Dataset | mean \+ var | mean |
> | :---: | :---: | :---: |
> | ImageNet-C | 66.5 | 60.6 |
>
> > **Comment 7:** How do you select the fixed threshold used to flag spikes in feature alignment.
>
> We use the prediction entropy for spike detection because it is a simple and direct indicator of prediction uncertainty. When a domain shift occurs, this uncertainty typically **spikes** due to the mismatch between source-trained decision boundaries and shifted inputs. The threshold is set to 1 in the paper based on our empirical analysis. Specifically, we analyzed the prediction entropy across batches and found that the entropy usually ranges from 0 to 0.8 in the training set. Therefore, we empirically set the threshold as 1\. To evaluate the impact of the threshold selection, we added additional experiments with different threshold values during the rebuttal. As shown in Table R1-6, the impact on accuracy is marginal. This is because (1) shift detection is mainly valid at the boundary of the domain transitions and (2) as long as the threshold can differentiate two domains, the performance will not be affected.
>
> Table R1-6. Impact of the shift detection threshold on accuracy (%).
>
> | Dataset | 0.5 | 1.0 | 1.5 | 2.0 |
> | :---: | :---: | :---: | :---: | :---: |
> | ImageNet-C | 66.4 | 66.5 | 66.4 | 66.4 |
> | CIFAR100-C | 77.0 | 77.0 | 77.1 | 76.8 |
>
> > **Comment 8 :** Can you ablate the choice of k views augmentation to see its effect on the main results ?
>
> We thank the reviewer for the suggestion. Our default setting in the paper is K=2 and during rebuttal, we added another ablation on the number of augmentation views from K=0 to 4 in the following table R1-7. We observe that multi-view augmentation consistently improves performance over K=0 (no augmentation), with the largest gain from K=0 to K=1, and a smaller but still clear gain from K=1 to  K=2. Beyond K=2, the improvements saturate and become marginal (\<0.1%). Since increasing K also increases compute and memory, we adopt **K=2** as the default to capture most of the benefit of augmentation while keeping the overhead modest. We will add the results to the final manuscript.
>
> Table R1-7. Impact of the number of test-time augmentation views K on top-1 accuracy (%).
>
> | Dataset | K \= 0 | K \= 1 | K \= 2 | K \= 3 | K \= 4 |
> | :---: | :---: | :---: | :---: | :---: | :---: |
> | ImageNet-C | 64.5 | 66.2 | 66.5 | 66.6 | 66.6 |
> | CIFAR100-C | 75.7 | 76.6 | 77.0 | 77.0 | 77.1 |

---

### Author Response · Authors · 2025-11-20
**A Common Response to the AC and All Reviewers**

We sincerely thank the AC and all reviewers for their time and constructive feedback. We first summarize several important strengths of our work **recognized by the reviewers**.

### Strengths highlighted by reviewers

- **Reviewer 1 (mDae​, ​Score: 10, Conf: 5)**
  Highlighted our *clear motivation and novelty*, *elegant solution*, *exceptional efficiency*, *strong empirical results*, and *generality (architecture-agnostic)*.

- **Reviewer 2 (bwXx​, ​Score: 6, Conf: 4)**
  Praised the *elegant embedding-space interpretation* and the practicality of our method for *edge devices*.

- **Reviewer 3 (9W85​, ​Score:  2, Conf: 3)**
  Acknowledged that the paper is *well written* and has *two interesting properties: backpropagation-free and architecture-agnostic*, and also highlighted that we *emphasize the method’s efficiency in practical settings*.

- **Reviewer 4 (Lsas​, ​Score: 4, Conf: 4)**
  Recognized our empirical findings, the *novel approach of PEA without model parameter updates*, and the *comprehensive ablation study* demonstrating the necessity of each component.

---

During the rebuttal, we conducted **extensive new experiments** and provided **important clarifications** to address all concerns, adding **13 new experimental studies with 13 new tables and 3 new figures**.

### New experiments and clarifications

- **Batch size 1/2 evaluation (R1, R2).**
  PEA achieves **61.6% / 69.5% at batch size 1** on ImageNet-C / CIFAR100-C while several baselines fail. We also show that EMA is crucial for stability in this regime (Tables R1-2, R1-3, R2-1, R2-2).

- **Mixed-domain experiments (R2, R4).**
  When all 15 domains are randomly shuffled within batches, PEA still achieves SOTA: **72.0%**, outperforming all baselines (Tables R2-5, R4-1).

- **Source data ablation (R1, R3).**
  Using only **10% of source samples** is sufficient for the offline computation, demonstrating the efficiency of our one-time preprocessing step (Tables R1-1, R3-1). Further, we implemented a detailed ablation using different models and datasets to show that only a few source data achieve robust statistics estimation (Table R3-D).

- **Architecture generality (R4).**
  ResNet-50 visualizations show that the same structural changes occur in CNNs, confirming that the phenomenon is caused by **domain shift** rather than architectural design (Figure 7 and Figure 8 in the revised paper).

- **Dataset diversity and generality (R3).**
  - To demonstrate that our method is not limited to corruption benchmarks, we added two new datasets, PACS and DomainNet, showing that PEA generalizes to more realistic domain shifts (Tables R3-C and R3-F).

- **Comprehensive ablations (All).**
  - Distance metric comparison showing that L2 is sufficient (Table R1-4).
  - Mean-only vs. mean+variance, showing that variance is crucial (Table R1-5).
  - Threshold sensitivity across \([0.5, 2.0]\), showing robustness (Tables R1-6, R2-3, R4-2).
  - Number of augmentation views, showing that \(K = 2\) is a good operating point (Tables R1-7, R2-4, R4-3).

- **Additional baselines and conceptual clarifications (R4, R2, R3).**
  - Added **2 new baselines** T3A (Table R4-4) and LAW (Table R3-E), with PEA outperforming T3A by **9.6%** and LAW by **3.6%**.
  - Clarified novelty relative to CORAL (explicit *test-time* embedding alignment vs. *training-time* model updates).
  - Provided a detailed FOA comparison (multi-layer covariance alignment vs. single-layer mean shift with only ~0.9% gain from activation shifting).
  - Distinguished PEA from LeanTTA (BN-specific vs. architecture-agnostic), ZOA (parameter updates vs. no parameter updates), and ADAPT (VLM-specific vs. general CNN/ViT classifiers).
  - Cited and connected to GALA and PALM for layer-wise adaptation.
  - Explained SPA evaluation protocol differences (per-domain vs. lifelong TTA).
  - Softened claims about geometric transformations to acknowledge non-linear and local effects.
  - Added **a new quantitative analysis** showing that intermediate embeddings under domain shift exhibit non-trivial components of translation, scaling, and rotation(Table R3-A).
---
We understand that, due to the recent security incident, reviewers were unable to participate during the final rebuttal period. As only **Reviewer 9W85** responded, we provided additional evaluations and clarifications, and the reviewer confirmed that our response **addressed the concerns on novelty and claim support**. To further respond to this reviewer’s remaining comments on datasets and baselines, we additionally included one more baseline and two new datasets in our rebuttal.

Overall, we believe that our rebuttal, including **13 new experimental studies with 13 new tables and 3 new figures**,  addresses all reviewer concerns. Furthermore, we believe that our method’s unique combination of being backprop-free, architecture-agnostic, and edge-deployable constitutes a significant and practical contribution.

---

### Meta-Review · Area_Chair_n6WE · 2025-12-28

**Summary:**

This paper presents a backpropagation-free and architecture-agnostic test-time adaptation method based on progressive embedding alignment. Reviewers highlighted the method’s strong efficiency, practical relevance, and applicability to both CNN and ViT models, particularly in resource-constrained settings. While initial reviews raised concerns about the novelty of the underlying assumptions, the strength of empirical support for the proposed geometric interpretation, and the scope of evaluation, the rebuttal and subsequent discussion provided substantial additional evidence and clarifications. Overall, the paper demonstrates a practically effective approach to test-time adaptation.
The initial reviewer scores were 10, 6, 4, and 2. The reviewer who assigned a rating of 2 engaged in a constructive discussion with the authors, and some of the key concerns were partially addressed. Based on these considerations, I recommend Accept.

**Reviewer Concerns:**

Most of the major concerns were addressed during the rebuttal and discussion phase. In particular, reviewers questioned whether interpreting domain shifts using simple embedding statistics was sufficiently novel and well supported, especially in light of prior work on distribution shift and OOD detection. In response, the authors clarified how their test-time adaptation setting differs from detection-focused approaches and provided additional quantitative evidence. The newly added reconstruction analyses show that simple statistic-based linear transformations can effectively approximate and mitigate the observed embedding shifts.
Concerns about the experimental scope were also largely resolved. Although the original submission mainly focused on synthetic corruption benchmarks, the authors added evaluations on. more realistic domain adaptation datasets. These results demonstrate that the proposed method remains effective beyond corruption-only settings. In addition, the dependence on source-domain statistics was examined more carefully through extended ablation studies across different datasets and architectures, showing that strong performance can be achieved even with a small fraction of source data.
Some limitations still remain. The geometric interpretation of domain shifts is a simplified view and may not fully capture more complex or non-linear effects, so a deeper theoretical characterization is left for future work.

**Reviewer Scores:**

This submission received split reviews, with initial scores of 10, 6, 4, and 2. After considering the rebuttal and the subsequent discussion, particularly the exchange with Reviewer 9W85 (initial rating: 2), the key concerns appear to have been sufficiently addressed, and I recommend Accept.

---

### Decision · Program_Chairs · 2026-01-26

Accept (Poster)